# EXPLORING THE POTENTIAL OF ENCODER-FREE ARCHITECTURES IN 3D LMMS

**Yiwen Tang**[1,2]__*__, **Zoey Guo**[3]__*__, **Zhuhao Wang**[4]__*__, **Ray Zhang**[3]__*__, **Qizhi Chen**[1], **Junli Liu**[1,2],
**Delin Qu**[1], **Dong Wang**[1], **Bin Zhao**[1,2][†], **Xuelong Li**[5]

[1]**Shanghai AI Laboratory** [2]**Northwestern Polytechnical University**
[3]**The Chinese University of Hong Kong** [4]**Tsinghua University** [5]**Tele AI**

## ABSTRACT

Encoder-free architectures have been preliminarily explored in the 2D Large Multimodal Models (LMMs), yet it remains an open question whether they can be effectively applied to 3D understanding scenarios. In this paper, we present the first comprehensive investigation into the potential of encoder-free architectures to alleviate the challenges of encoder-based 3D LMMs. These long-standing challenges include the failure to adapt to varying point cloud resolutions during inference and the point features from the encoder not meeting the semantic needs of Large Language Models (LLMs). We identify key aspects for 3D LMMs to remove the pre-trained encoder and enable the LLM to assume the role of the 3D encoder: 1) We propose the LLM-embedded Semantic Encoding strategy in the pre-training stage, exploring the effects of various point cloud self-supervised losses. And we present the Hybrid Semantic Loss to extract high-level semantics. 2) We introduce the Hierarchical Geometry Aggregation strategy in the instruction tuning stage. This incorporates inductive bias into the LLM layers to focus on the local details of the point clouds. To the end, we present the first Encoder-free 3D LMM, ENEL. Our 7B model rivals the state-of-the-art model, PointLLM-PiSA-13B, achieving 57.91%, 61.0%, and 55.20% on the classification, captioning, and VQA tasks, respectively. Our results show that the encoder-free architecture is highly promising for replacing encoder-based architectures in the field of 3D understanding. The code is released at `https://github.com/Ivan-Tang-3D/ENEL`.

## 1 INTRODUCTION

Large Language Models (LLMs) Touvron et al. (2023); Bai et al. (2023) have gained unprecedented attention for their proficiency in understanding and generating complex language scenarios. Building upon these advances, many recent efforts have been made to develop Large Multimodal Models (LMMs), empowering LLMs with the capability to interpret multimodal information, such as 2D images Li et al. (2024); Jing et al. (2025); Tang et al. (2025b); Yang et al. (2025), 3D point clouds Guo et al. (2023a); Xu et al. (2025); Wang et al. (2025); Guo et al. (2023b); Tang et al. (2024a;b; 2025a); Chen et al. (2024b) and visual generation Tong et al. (2025); Jiang et al. (2025); Guo et al. (2025d;c;b).

Mainstream LMMs are typically encoder-based, relying on heavyweight yet powerful pre-trained encoders (e.g., CLIP Radford et al. (2021) for 2D and I2P-MAE Zhang et al. (2023a) for 3D). While these pre-trained encoders offer robust multimodal embeddings enriched with pre-existing knowledge, they also introduce challenges that could limit the future advancement of multimodal understanding. To mitigate the limitations introduced by visual encoders in VLMs—such as resolution, aspect ratio, and semantic priors—many encoder-free LMM studies Li et al. (2025a); Diao et al. (2024a; 2025); Lei et al. (2025); Luo et al. (2025) have explored the possibility of training without pre-trained encoders.

---

*Equal Contribution.
[†]Corresponding Author.

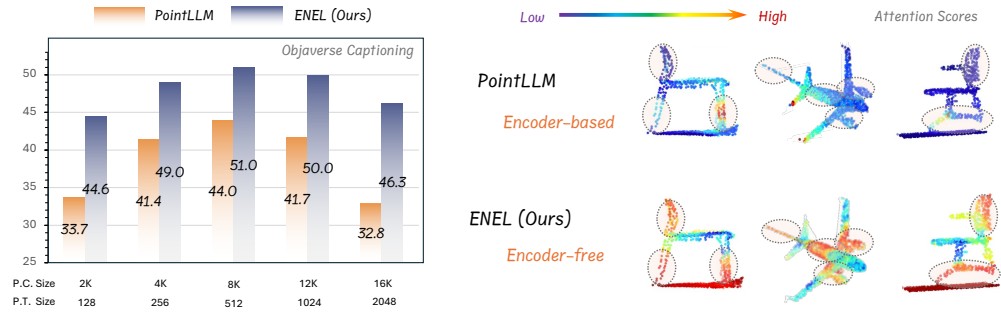

(a) Point Cloud Resolution Limitation      (b) Embedding Semantic Discrepancy

Figure 1: **Issues of encoder-based 3D LMMs.** (a) **Point Cloud Resolution Limitation.** During training, the point cloud size (P.C. Size) and point token size (P.T. Size) are fixed at 8192 and 512, respectively. And we adjust these two sizes during inference, point cloud size from 2K to 16K and the corresponding point token size from 128 to 2048. We evaluate them on the captioning task of the Objaverse benchmark using GPT-4 score as the evaluation metric. (b) **Embedding Semantic Discrepancy.** We visualize the attention scores of the average text token to the point tokens, where **red indicates higher values.** The point tokens in the encoder-free architecture exhibit stronger textual semantic relevance needed for the LLM.

Specifically for 3D LMMs, the encoder-based architecture has the following potential drawbacks: (1) *Point Cloud Resolution Limitation.* 3D encoders are often pre-trained on point cloud data at a fixed resolution, such as 8,192 points for Point-BERT Yu et al. (2022) in PointLLM Xu et al. (2025). However, during inference, the resolution of point clouds may vary (e.g., 12,000 or 4,000 points). This difference between training and inference resolutions can result in the loss of spatial information when extracting 3D embeddings, leading to difficulties for LLMs to comprehend, as showcased in Figure 1 (a). (2) *Embedding Semantic Discrepancy.* 3D encoders are typically pre-trained using self-supervised methods like MAE Pang et al. (2022); Tang et al. (2024a;b) and contrastive learning Xie et al. (2020); Qi et al. (2023), but these training objectives may not align with the specific semantic needs of LLMs. In other words, they may not capture the most relevant semantics for LLMs to understand 3D objects, as visualized in Figure 1 (b). Even when a projection layer is used to connect 3D encoders with LLMs, simple MLPs are often insufficient for a complete semantic transformation. Given these issues, we ask: *Is it possible to explore an encoder-free architecture for 3D LMMs, eliminating the 3D encoder and instead integrating its functionality directly within the LLM itself?*

In this paper, we present the first systematic investigation into the potential of an encoder-free architecture for 3D LMMs. To minimize external influences and ensure clarity, we use the pioneering and sufficiently concise PointLLM Xu et al. (2025) as our encoder-based baseline, which consists of two progressive training stages: pre-training and instruction tuning. We evaluate the performance on 3D classification Deitke et al. (2023), 3D captioning Deitke et al. (2023) and 3D VQA Deitke et al. (2023) tasks. Specifically, to remove the encoder while mitigating any performance degradation, we explore solutions to the following two key questions:

*(1) How can we compensate for the high-level 3D semantics originally extracted by the 3D encoder?* In 3D LMMs, the raw point cloud input is first passed through a token embedding module for low-level tokenization, before being processed by the main 3D encoder, usually a Transformer Vaswani (2017), to generate high-level embeddings. Skipping the encoder entirely poses a challenge in capturing the complex spatial structures of 3D point clouds. To address this, we propose a strategy called **LLM-embedded Semantic Encoding** in the pre-training stage. First, we adopt a simple yet effective token embedding module that captures as much informative semantic content as possible. These 3D tokens are then directly fed into the LLM. Next, we aim to shift the responsibility of capturing high-level 3D semantics to the LLM itself. To guide this process, we explore various 3D self-supervised loss functions, such as masked modeling loss and distillation loss, and ultimately propose the Hybrid Semantic Loss as the most effective choice. Further, we make the early layers of the LLM to be learnable, allowing them to specialize in multimodal alignment.

*(2) How can we integrate inductive bias into LLMs for better perception of 3D geometric structures?* Pre-trained 3D encoders typically embed explicit inductive bias into their architectures to progressively capture multi-level 3D geometries. For instance, models like Point-M2AE Zhang et al. (2022) use a local-to-global hierarchy, which is a concept also common in convolutional layers for 2D image processing He et al. (2016). In contrast, LLMs employ standard Transformer architectures,

where each layer processes the same number of tokens, representing the same semantic level across the network. In the absence of the encoder, we introduce the approach of **Hierarchical Geometry Aggregation** during the fine-tuning stage. In the early layers of the LLM, we aggregate 3D tokens based on their geometric distribution using Dynamic Grid Sampling. This approach enables the LLM to gradually integrate detailed 3D semantics and develop a more holistic understanding of the 3D object. In the later layers, we reverse this aggregation, propagating the tokens back to their original distribution to maintain the fine-grained representation necessary for complex tasks.

Through a series of experimental investigations, we have uncovered the strong potential of applying encoder-free architecture to the 3D LMM domain. Building on our insights, we introduce ENEL, an **EN**coder-fre**E** 3D **L**MM evolved from Vicuna-7B Chiang et al. (2023) using the same training dataset from PointLLM. Notably, without any 3D encoders, ENEL-7B achieves comparable performance to the state-of-the-art PointLLM-PiSA-13B Guo et al. (2025a). We hope ENEL may provide the community with an effective path for adapting the encoder-free architecture to 3D scenarios.

Our main contributions are summarized as follows:

• We present the first comprehensive empirical study of applying encoder-free architectures to the 3D LMM domain, offering valuable insights for the field.

• We aim to transfer the original roles of 3D encoders to the LLM itself, and propose the LLM-embedded Semantic Encoding and Hierarchical Geometry Aggregation strategy, both of which have been validated as effective.

• We further introduce ENEL, a concise and well-performed encoder-free 3D LMM, which, at the 7B parameter scale, achieves 57.91%, 61.0%, and 55.20% on 3D captioning, classification, and 3D VQA tasks, respectively, on par with existing encoder-based models.

## 2 RELATED WORK

**3D LMM.** Recent advancements in integrating large language models (LLMs) with 3D data have led to significant progress in both object-level and scene-level understanding. At the object level, early approaches like Hong et al. (2024) utilize 2D rendering to leverage 2D LLMs, but this sacrifices geometric details. More recent models, including Point-Bind LLM Guo et al. (2023a), PointLLM Xu et al. (2023b) and ShapeLLM Qi et al. (2024), directly encode point clouds and align them with LLMs, by combining the 3D encoder with a powerful language model, effectively fusing geometric, appearance, and linguistic information. MiniGPT-3D Tang et al. (2024c) is introduced, which efficiently aligns 3D point clouds with LLMs by leveraging 2D priors from 2D-LLMs. It employs a four-stage cascaded training strategy along with a Mixture of Query Experts (MoQE) module. Zeng et al. propose GreenPLM Tang et al. (2025c), an energy-efficient framework that directly translates monolingual pre-trained language models into other languages using bilingual lexicons. At the scene level, models like Chat-3D Wang et al. (2023) and Scene-LLM Fu et al. (2024) focus on understanding complex spatial relationships through dialogue and tasks like captioning. Scene-LLM Fu et al. (2024) enhances embodied agents' abilities in interactive 3D indoor environments by integrating both scene-level and egocentric 3D information. Grounded 3D-LLM Chen et al. (2024d) utilizes referent tokens to reference specific objects within 3D scenes, enabling tasks such as object detection and language grounding. However, conventional encoder-based 3D LMMs commonly suffer from limitations, specifically Point Cloud Resolution Limitation and Embedding Semantic Discrepancy, which stem from the inductive bias inherent in the 3D pre-trained encoder. Our ENEL alleviates these restrictions by removing the encoder and utilizes a lightweight architecture to significantly boost performance.

**Encoder-free Vision-Language Models.** Traditional vision-language models (VLMs) often rely on vision encoders to extract visual features before processing them with language models, integrating image encoders like CLIP Radford et al. (2021) and DINO V2 Oquab et al. (2023). However, recent efforts have explored encoder-free VLMs for their simplicity. Approaches like ChameleonTeam (2024); Xie et al. (2024) use VQ tokenizers Esser et al. (2021) or linear projection layers Diao et al. (2024a); Chen et al. (2024c) to represent images. Fuyu-8B Bavishi et al. (2023), a pure decoder-only model, directly processes image patches through linear projections, handling high-resolution images but showing only average performance. The EVE series Diao et al. (2024b; 2025) eliminates the need for a separate vision encoder by bridging vision-language representation within a unified decoder and enhancing visual recognition capabilities through additional supervision. Mono-InternVL series Luo

Table 1: **Token Embedding.** Performance on Objaverse with PointLLM-7B as the baseline. 'Cls'/'Cap': classification/captioning tasks. 'Avg': accuracy under prompts *"What is this?"* and *"This is an object of."* 'S-BERT': Sentence-BERT. 'T.E.': our designed token embedding module.

| Method | Cls (Avg) | Cap | |
|---|---|---|---|
| | GPT-4 | GPT-4 | S-BERT |
| PointLLM-7B | 53.00 | 44.85 | 47.47 |
| - Encoder | 35.50 | 33.37 | 41.19 |
| + 2-layer T.E. | 40.60 | 38.85 | 43.25 |
| **+ 3-layer T.E.** | **45.55** | **41.36** | **44.82** |
| + 4-layer T.E. | 43.00 | 40.47 | 43.50 |

Table 2: **Learnable Layers.** We set the LLM early layers to be learnable. 'LR' represents the learning rate during the pre-training stage, with the original learning rate set to 2e-3.

| Method | LR | Cls (Avg) | Cap | |
|---|---|---|---|---|
| | | GPT-4 | GPT-4 | S-BERT |
| PointLLM-7B | 2e-3 | 53.00 | 44.85 | 47.47 |
| + 2 learnable layers | 2e-3 | 40.00 | 40.20 | 44.82 |
| | 4e-4 | 44.00 | 42.62 | 46.30 |
| **+ 4 learnable layers** | 2e-3 | 43.75 | 40.13 | 45.76 |
| | **4e-4** | **47.90** | **43.50** | **46.70** |
| + 8 learnable layers | 2e-3 | 42.35 | 37.91 | 41.28 |
| | 4e-4 | 46.70 | 42.80 | 46.14 |
| + 12 learnable layers | 2e-3 | 41.55 | 40.05 | 41.40 |
| | 4e-4 | 46.15 | 42.39 | 46.00 |

et al. (2024; 2025) leverage visual experts and progressive visual pre-training (EViP/EViP++) to achieve stable optimization and competitive performance. SAIL series Lei et al. (2025) directly encode raw pixels and decodes language within a single architecture, achieving competitive vision-language performance without pre-trained vision encoders. The key idea behind ENEL is enabling the LLM to assume the functionality of the encoder by effective and efficient methods. This approach diverges from 2D encoder-free LMMs, which tend to focus on larger datasets and more complex structures for better results.

# 3 INVESTIGATION OF ENCODER-FREE 3D LMM

## 3.1 PRELIMINARY

***Encoder-free in 2D LMMs.*** ELVA Li et al. (2025a) is an encoder-free Video-LLM that directly models nuanced video-language interactions without relying on a vision encoder. EVE Diao et al. (2024a) and its successor EVEv2 Diao et al. (2025) are designed as efficient encoder-free vision-language models. SAIL Lei et al. (2025) serves as a unified transformer for vision and language, while Mono-InternVL Luo et al. (2025) represents a monolithic multimodal LLM. In parallel, Fuyu-8B Bavishi et al. (2023), a decoder-only transformer developed by Adept AI, has gained substantial community adoption. A common characteristic across these works is the adoption of a lightweight, randomly initialized token embedding layer to convert inputs into tokens for the LLM. This design eliminates the need for a dedicated vision encoder and enables end-to-end training and inference.

***Pre-trained Encoders in 3D LMMs.*** Traditionally, 3D pre-trained encoders are characterized by two properties: (1) independent pretraining on point cloud tasks (e.g., reconstruction), and (2) structural decoupling, where they are connected to the language model through projection layers. In 3D LMMs, commonly adopted encoders refer to pre-trained models such as PointMAE Pang et al. (2022), PointBERT Yu et al. (2022), and Uni3D Zhou et al. (2023).

***Overall Architecture.*** We select PointLLM as the baseline model for the exploration and evaluate the performance of different strategies on the Objaverse dataset Deitke et al. (2023), using GPT-4 scores combined with traditional metrics as our evaluation metrics. **Point Embedding Layer.** As shown in Figure 2, we first remove the encoder of PointLLM and adopt the original token embedding Yu et al. (2022). However, the coarse structural design results in a significant performance degradation, as observed in Table 1, where the GPT-4 scores for the classification and captioning tasks decrease by 17.5% and 10.48%, respectively. To mitigate excessive information loss and provide refined local features to the LLM, we adopt a small network with a limited number of parameters, which is a lightweight variant of Point-PN Zhang et al. (2023b). Specifically, for the input $\{P_i\}_{i=1}^N$, we apply Farthest Point Sampling (FPS) for downsampling the number of points, k-Nearest Neighbors (k-NN) with group size k for local aggregation, and learnable linear layers for feature encoding. After a series of repetitive operations and the projection layer, we transform the point clouds into high-dimensional vectors $\{F_i\}_{i=1}^M \in \mathbb{R}^{M \times D_1}$. In Table 1, we experiment with token embedding at different depths and find that three layers yield the best performance. **3D Encoding & Alignment.** We discover that the absence of the encoder results in a lack of context modeling in point cloud feature processing. Therefore, we attempt to have the early layers of the LLM take on the encoder's role in capturing

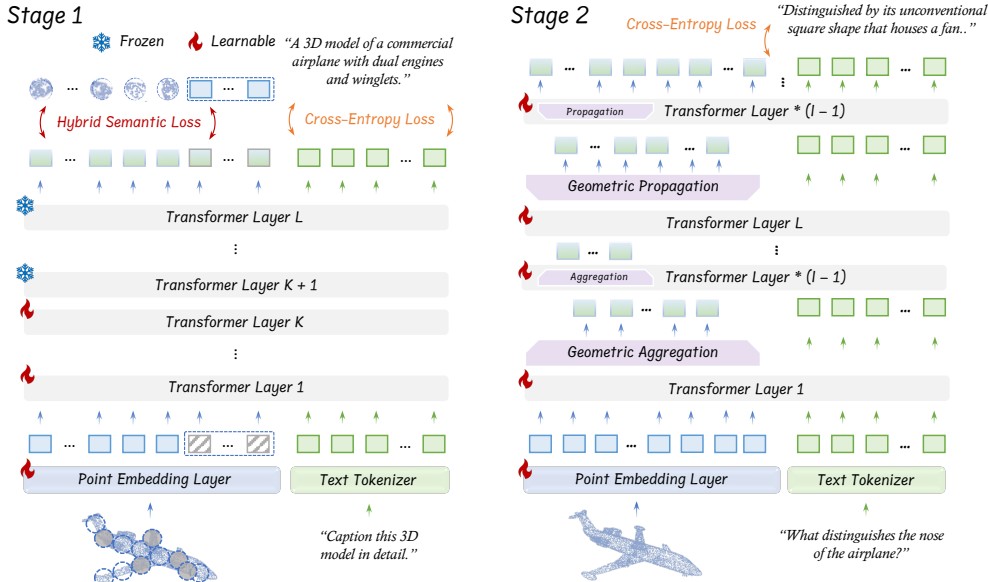

Figure 2: **Overall Pipeline of ENEL.** The training is divided into two stages: the pre-training stage and the instruction tuning stage. In the first stage, we set the first $K$ layers to be learnable and apply the proposed Hybrid Semantic Loss to embed high-level semantics into the LLM. In the second stage, we adopt the Hierarchical Geometric Aggregation strategy to capture local structures of point clouds.

global interactions of features, further encoding the point cloud features. In the pre-training stage, we set the first K layers of the frozen LLM to be learnable. Within the shared semantic space, 3D tokens and text tokens interact and align naturally. Early Fusion provides a more practical way to achieve modality alignment between 3D and textual semantic spaces. Meanwhile, we experiment with different learning rates. As shown in Table 2, a smaller learning rate yields better results by stabilizing early layer optimization. Based on the designed token embedding module, setting the first four layers to be learnable yields the best results.

## 3.2 LLM-EMBEDDED SEMANTIC ENCODING

The lack of the 3D encoder results in insufficient encoding of point cloud semantic information, which greatly hinders the LLM to understand the structural details of point clouds. Most existing 3D encoders use self-supervised losses to embed the high-level semantics of point clouds into the transformer, primarily categorized into four types: Masked Modeling Loss Pang et al. (2022), Reconstruction Loss Qi et al. (2023), Contrastive Loss Khosla et al. (2020); Qu et al. (2025), and Knowledge Distillation Loss Zhang et al. (2023a). Based on the proposed token embedding module and LLM learnable early layers, we implement and evaluate the effects of these losses on the encoder-free 3D LMM in the pre-training stage, as described in Figure 3. Finally, we propose the Hybrid Semantic Loss, which assists the LLM to learn the relationship between local spatial information in the point clouds and grasp the high-level 3D semantics.

**Masked Modeling Loss.** In the pre-training stage, we apply the Masked Modeling Loss to the point tokens processed by the LLM, as shown in Figure 3 (a). Through the token embedding module, the point clouds $\{P_i\}_{i=1}^{N}$ are divided into point patches $\{G_i\}_{i=1}^{M} \in \mathbb{R}^{M \times k \times 3}$ and the corresponding point tokens $\{F_i\}_{i=1}^{M}$. We randomly mask the point tokens with a masking ratio $r$, and replace them with learnable tokens. The masked feature tokens can be denoted as $\{F_{\text{gt}_i}\}_{i=1}^{M*r}$, which serve as the ground truth for the loss computation. After the masked tokens are replaced with learnable tokens and processed by the LLM, a linear layer is applied to predict the point tokens $\{F_{\text{pre}_i}\}_{i=1}^{M*r} \in \mathbb{R}^{M*r \times D_1}$, and the Mean Squared Error (MSE) is computed between $F_{\text{pre}}$ and $F_{\text{gt}}$. The optimization is:

$$\mathcal{L}_{\text{mask}} = \frac{1}{M*r} \sum_{i=1}^{M*r} \left( \|F_{\text{pre}_i} - F_{\text{gt}_i}\|_2^2 \right).$$ (1)

The specific process of applying Masked Modeling to point patches $G$ is detailed in Appendix A.2.1.

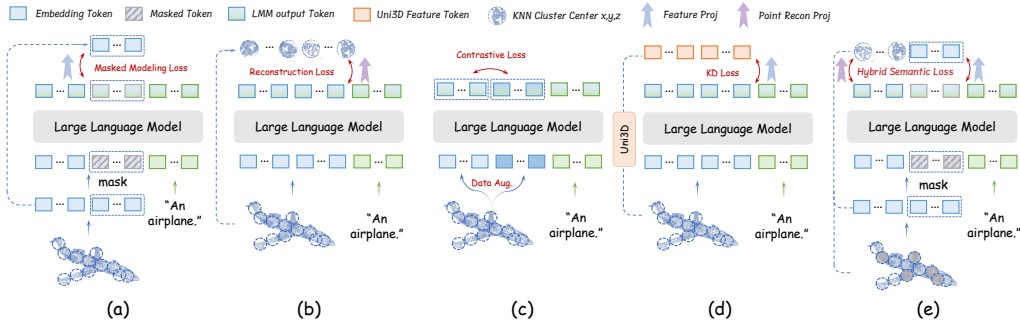

Figure 3: **Point Cloud Self-Supervised Learning Losses.** In the pre-training stage, we explore common self-supervised learning losses for the encoder-free 3D LMM: (a) Masked Modeling Loss, (b) Reconstruction Loss, (c) Contrastive Loss, and (d) Knowledge Distillation Loss. The (e) represents our proposed Hybrid Semantic Loss, specifically designed for the encoder-free architecture.

**Reconstruction Loss.** After the point feature tokens $\{F_i\}_{i=1}^{M}$ are encoded by the LLM, the tokens are transformed to the point patches $\{G_{\text{pre}_i}\}_{i=1}^{M} \in \mathbb{R}^{M \times k \times 3}$ through a linear layer. We utilize the $l_2$ chamfer distance to align the predicted $G_{\text{pre}}$ with the ground truth $G$, reconstructing the original spatial information, as illustrated in Figure 3 (b). This approach encourages the LLM to learn the high-level semantics of the point cloud while preserving the critical structure and key features of the point cloud input. The optimization target $L_{\text{recon}}$ can be written as

$$\frac{1}{M} \sum_{i=1}^{M} \left( \min_{j} \|a_i - b_j\|_2^2 + \min_{j} \|b_i - a_j\|_2^2 \right), \tag{2}$$

where $a = G_{\text{pre}}$, $b = G$. The procedure for reconstructing feature F is detailed in Appendix A.2.1.

**Contrastive Loss.** We conduct contrastive learning Khosla et al. (2020) at the point cloud level, where we contrast two transformed versions of the point cloud in the Figure 3 (c). Given a sampled point cloud $\{P_i\}_{i=1}^{N}$, we apply two random geometric transformations $T_1$ and $T_2$, including rotation and translation, to obtain $P_{T1}$ and $P_{T2}$. The two augmented point clouds are separately paired with the original text query and processed through the LLM to obtain their respective feature tokens $F_{T1} \in \mathbb{R}^{M \times D_1}$ and $F_{T2} \in \mathbb{R}^{M \times D_1}$. Within the mini-batch, the two feature tokens derived from the same point cloud serve as positive pairs, while they are considered negative pairs with other point clouds. Using NCESoftmaxLoss, we aim to maximize the similarity of positive pairs and minimize the similarity of negative pairs, encouraging the LLM to learn geometric equivariance of point clouds. The $\mathcal{L}_{\text{contrast}}$ is shown as below, where B stands for the training batch size.

$$\frac{1}{B} \sum_{i=1}^{B} \left( -\log \frac{\exp(\mathbf{F}_{T1_i} \cdot \mathbf{F}_{T2_i}/\tau)}{\sum_{j=1}^{B} \exp(\mathbf{F}_{T1_i} \cdot \mathbf{F}_{T2_j}/\tau)} \right). \tag{3}$$

**Knowledge Distillation Loss.** We select the powerful Uni3D-L Zhou et al. (2023) as the teacher encoder, input the point cloud into the 3D encoder, and obtain the output feature $F_{\text{teacher}} \in \mathbb{R}^{M \times D_2}$. The Mean Squared Error (MSE) between the LLM output tokens $F_{\text{student}}$ and $F_{\text{teacher}}$ is computed to align $F_{\text{student}}$ as closely as possible to $F_{\text{teacher}}$, thereby transferring the knowledge embedded in the 3D encoder to the LLM. By obtaining additional supervision from the Uni3D, the LLM better captures the complex structures in the point cloud data, as displayed in Figure 3 (d). The objective function is:

$$\mathcal{L}_{\text{KD}} = \frac{1}{M} \sum_{i=1}^{M} \left( \|F_{\text{student}_i} - F_{\text{teacher}_i}\|_2^2 \right). \tag{4}$$

**Experiments and Insights.** As shown in Table 3, we compare the effects of common self-supervised learning losses in the pre-training stage, where they are summed with the LLM cross-entropy loss Touvron et al. (2023), each with a coefficient of 1. The observations are summarized as below:

- **The point cloud self-supervised losses generally benefit the encoder-free 3D LMM.** Compared to previous experimental results, where the GPT scores for the classification and captioning tasks are 47.90% and 43.50%, the self-supervised losses bring about the significant improvements. This

Table 3: **LLM-embedded Semantic Encoding.** In pre-training, we explore the effects of different self-supervised learning losses targeting point tokens. $\Psi$ and $\Phi$ denote mask ratios of 60% and 30%, respectively. Subscripts $patch$ and $feat$ indicate loss targets. For Hybrid Semantic Loss, the subscripts $patch$ and $feat$ refer to the masked modeling target, with reconstruction targeting the corresponding $feat$ and $patch$.

| Method | Cls (Avg) | Cap | |
|---|---|---|---|
| | GPT-4 | GPT-4 | S-BERT |
| PointLLM-7B | 53.00 | 44.85 | 47.47 |
| Masked Modeling Loss$_{patch}^{\Psi}$ | 47.00 | 43.64 | 45.36 |
| Masked Modeling Loss$_{patch}^{\Phi}$ | 49.00 | 45.20 | 46.29 |
| Masked Modeling Loss$_{feat}^{\Psi}$ | 48.50 | 43.90 | 45.30 |
| Masked Modeling Loss$_{feat}^{\Phi}$ | 48.50 | 45.85 | 46.93 |
| Reconstruction Loss$_{patch}$ | 48.00 | 45.56 | 46.33 |
| Reconstruction Loss$_{feat}$ | 47.50 | 44.05 | 46.18 |
| Contrastive Loss | 42.50 | 41.21 | 43.77 |
| Knowledge Distillation Loss | 48.00 | 43.87 | 46.09 |
| Hybrid Semantic Loss$_{patch}$ | 50.00 | 45.24 | 46.59 |
| **Hybrid Semantic Loss$_{feat}$** | **52.00** | **47.65** | **47.30** |

is because the self-supervised learning loss forces transformations on the complex point clouds through certain task design. This encourages the LLM to not simply memorize specific point cloud data but to learn the underlying geometric relationships and high-level semantic information.

- **Among the self-supervised learning losses, the Masked Modeling Loss demonstrates the strongest performance improvement.** It achieves GPT-4 scores of 48.5% and 45.85% for classification and captioning tasks, respectively. The application of the masked modeling to the point features facilitates the embedding of high-level semantics from point clouds into the LLM. However, a higher mask ratio increases training difficulty, with 60% performing worse than 30%. In addition, explicitly reconstructing point patches helps capture complex structures and critical details in point clouds. Knowledge Distillation Loss falls short compared to the first two losses. Finally, Contrastive Loss, which fails to extract the detailed semantics, achieves the lowest performance.

**Hybrid Semantic Loss.** Based on the experimental results above, we propose the self-supervised learning loss specifically designed for the encoder-free 3D LMM—Hybrid Semantic Loss, as show-cased in Figure 3 (e). We apply a masking ratio $r$ to randomly mask point tokens from the token embedding. The masked tokens and the corresponding patches are referred to as $\{F_{\text{mask}_i}\}_{i=1}^{M*r}$ and $\{G_{\text{mask}_i}\}_{i=1}^{M*r}$, respectively. The remaining tokens are denoted as $\{F_{\text{vis}_i}\}_{i=1}^{M*(1-r)}$ and $\{G_{\text{vis}_i}\}_{i=1}^{M*(1-r)}$. Considering the autoregressive nature of the LLM and the unordered attribute of point clouds, we directly concatenate learnable tokens $\{F_{\text{learn}_i}\}_{i=1}^{M*r}$ to the end of $F_{\text{vis}}$, replacing the masked tokens. For the masked portion, we adopt masked modeling, and for the visible portion, we use the reconstruction strategy. After passing point tokens through the LLM, we compute the MSE between $F_{\text{learn}}$ and $F_{\text{mask}}$. The visible features $F_{\text{vis}}$ are transformed into $G_{\text{pred}}$, and the $L_2$ Chamfer distance is computed between $G_{\text{pred}}$ and $G_{\text{vis}}$. These two are added to the original cross-entropy loss with coefficients all equal to 1. This approach not only embeds high-level semantics into the LLM but also ensures geometric consistency throughout the point cloud learning process. With a 30% mask ratio and per-layer positional encoding of point tokens, it achieves 52.00% and 47.65% on the classification and captioning tasks, respectively. The inverse modeling process is described in Appendix A.2.1.

Our motivation arises from the observation that complex objectives, such as KD and contrastive learning, impose significant computational overhead yet often yield marginal gains compared to intrinsic data modeling losses like masked modeling. To address this, we propose the Hybrid Semantic Loss, which resolves the structural mismatch between 3D data and LLMs by exploiting two key properties: (1) the permutation invariance of point clouds, allowing learnable tokens to be appended after visible tokens without positional restoration; and (2) the encoder-free architecture, where 3D tokens are integrated into a causally-masked LLM instead of a bidirectionally-masked 3D encoder, fundamentally altering information flow between visible and masked tokens, enabling visible tokens to learn harder objectives while learnable tokens focus on lightweight reconstruction.

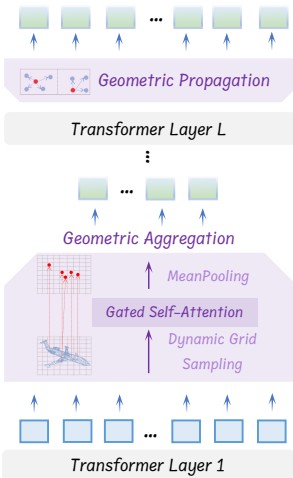

Figure 4: **Hierarchical Geometry Aggregation Strategy.** In the instruction tuning stage, we apply aggregation and propagation operations to the point tokens to capture the local structural details.

Table 4: **Hierarchical Geometry Aggregation.** In the instruction tuning stage, we conduct the experiments of Hierarchical Geometry Aggregation strategy. $l$ represents the number of aggregation and propagation operations. $H$ refers to the LLM layers between $l$ aggregation and $l$ propagation operations. + Self-Attn. represents the incorporation of the gated self-attention in the aggregation.

| Method | Cls (Avg) | Cap | |
|---|---|---|---|
| | GPT-4 | GPT-4 | S-BERT |
| PointLLM-7B | 53.00 | 44.85 | 47.47 |
| $l$=1 | 52.50 | 48.70 | 48.07 |
| $l$=2 | 51.00 | 46.67 | 48.12 |
| $l$=3 | 53.00 | 48.93 | 48.06 |
| $l$=4 | 45.00 | 45.48 | 46.90 |
| $H$=2 | 54.25 | 49.56 | 48.52 |
| $H$=4 | 52.50 | 48.61 | 47.81 |
| $H$=8 | 52.25 | 48.95 | 47.90 |
| + Self-Attn. | **55.55** | **51.03** | **48.79** |

## 3.3 HIERARCHICAL GEOMETRY AGGREGATION

3D encoders are designed with specific structures tailored for point clouds, such as local-to-global hierarchy Zhang et al. (2022) for exploring the geometric structure of the point cloud. However, in encoder-free architectures, the LLM itself does not have an explicit local modeling module. The self-attention mechanism is intended for modeling global interactions. Therefore, building upon the proposed Hybrid Semantic Loss, we explore in the instruction tuning stage how to enable the LLM to actively perceive 3D local details and complement the learned global semantics. To this end, we propose the Hierarchical Geometry Aggregation strategy.

**Implementation Details.** As depicted in Figure 4, from the LLM second layer, the input point tokens $\{F_{\text{input}_i}\}_{i=1}^{M}$, based on their corresponding coordinates $\{P_{\text{input}_i}\}_{i=1}^{M}$, are grouped by Dynamic Grid Sampling. The grid size follows a cumulative scaling strategy across aggregation layers. At the $i$-th aggregation layer, the grid size is:

$$s_i = \alpha \cdot e^{\sum_{j=1}^{i} \beta_j}, \qquad \beta_j = \gamma \cdot \tanh(\theta_j) + \beta_{\text{ctr}}, \tag{5}$$

where $\alpha = 0.02\,\text{m}$ and $s_i \in [s_{\min}, s_{\max}] = [0.02, 1]\,\text{m}$. To ensure the cumulative scaling stays within bounds across $l$ aggregation layers, we set:

$$\gamma = \frac{\ln\left(\frac{s_{\max}}{\alpha}\right) - \ln\left(\frac{s_{\min}}{\alpha}\right)}{2l}, \qquad \beta_{\text{ctr}} = \frac{\ln\left(\frac{s_{\max}}{\alpha}\right) + \ln\left(\frac{s_{\min}}{\alpha}\right)}{2l}, \tag{6}$$

where $l$ is the total number of aggregation layers. Each $\theta_j$ is randomly initialized from a standard normal distribution. Points within the same grid cell form local neighbors, with the set of all neighbors denoted as $\mathcal{G}_i$ having cardinality $M_i$. The neighborhood features $F_{\text{input}}^n \in \mathbb{R}^{M_i \times k \times D_1}$ are then collected, where $k$ denotes the maximum number of points across all cells. To handle varying point numbers across grid cells, we employ a padding strategy: for cells with fewer than $k$ points, we compute the mean-pooled feature of existing points and concatenate it repeatedly until reaching $k$ points per cell. For $F_{\text{input}}^n$, we employ the gated self-attention mechanism for intra-group interactions, grasping the local geometric structure. We multiply the self-attention output by a learnable parameter initialized from zero to adaptively adjust the required knowledge. We formulate it as

$$F_{\text{input}}^n{}' = tanh(\alpha) * \text{Self-Attn.}(F_{\text{input}}^n) + F_{\text{input}}^n. \tag{7}$$

On top of this, we apply pooling to fuse the features $F_{\text{input}}^n{}'$ within each neighbor, yielding aggregated tokens $\{F_{\text{agg}_j}^i\}_{j=1}^{M_i}$, formulated as

$$F_{\text{agg}}^i = \text{MeanPooling}(F_{\text{input}}^n{}'). \tag{8}$$

Table 5: **Comparison of different models on various 3D understanding tasks.** A primary focus is placed on GPT-4 evaluation, along with data-driven metrics (Sentence-BERT). The * indicates the Qwen2.5-7B LLM base and the ShapeLLM training data. The $\alpha$ denotes reproduced results. [†] denotes the model is implemented based on the ShapeLLM baseline.

| Model | Cap | | | | | | Cls (Avg) | QA |
|---|---|---|---|---|---|---|---|---|
| | GPT-4 | Sentence-BERT | SimCSE | BLEU-1 | ROUGE-L | METEOR | GPT-4 | GPT-4 |
| InstructBLIP-7BDai et al. (2023) | 45.34 | 47.41 | 48.48 | 4.27 | 8.28 | 12.99 | 43.50 | – |
| InstructBLIP-13BDai et al. (2023) | 44.97 | 45.90 | 48.86 | 4.65 | 8.85 | 13.23 | 34.25 | – |
| LLaVA-7BLiu et al. (2024) | 46.71 | 45.61 | 47.10 | 3.64 | 7.70 | 12.14 | 50.00 | – |
| LLaVA-13BLiu et al. (2024) | 38.28 | 46.37 | 45.90 | 4.02 | 8.15 | 12.58 | 51.75 | 47.90 |
| PointGPTChen et al. (2023) | – | – | – | – | – | – | 11.60 | – |
| Uni3DZhou et al. (2023) | – | – | – | – | – | – | 47.20 | – |
| 3D-LLMHong et al. (2023) | 33.42 | 44.48 | 43.68 | 16.91 | 19.48 | 19.73 | 45.25 | – |
| PointLLM-7BXu et al. (2023b) | 44.85 | 47.47 | 48.55 | 3.87 | 7.30 | 11.92 | 53.00 | 41.20 |
| PointLLM-13BXu et al. (2023b) | 48.15 | 47.91 | 49.12 | 3.83 | 7.23 | 12.26 | 54.00 | 46.60 |
| ShapeLLM-7BQi et al. (2024) | 46.92 | 48.20 | 49.23 | – | – | – | 54.50 | 47.40 |
| ShapeLLM-13BQi et al. (2024) | 48.94 | 48.52 | 49.98 | – | – | – | 54.00 | 53.10 |
| MiniGPT-3D$^\alpha$ Tang et al. (2024c) | 52.49 | 48.73 | 49.26 | – | – | – | 54.50 | 43.60 |
| PointLLM-PiSA-7BGuo et al. (2025a) | 48.63 | 48.47 | 49.08 | 3.80 | 7.25 | 12.38 | 54.50 | 42.90 |
| PointLLM-PiSA-13BGuo et al. (2025a) | 50.52 | 48.60 | 49.64 | 3.75 | 7.84 | 12.56 | 55.00 | 46.80 |
| **ENEL-7B** | 51.03 | 48.79 | 49.52 | 3.91 | 7.20 | 12.68 | 55.55 | 43.80 |
| **ENEL-7B**[†] | 53.26 | 48.75 | 49.94 | - | - | - | 56.00 | 48.90 |
| **ENEL-13B** | 53.24 | 48.92 | 50.17 | 3.72 | 7.89 | 12.31 | 56.00 | 48.50 |
| **ENEL-13B**[†] | 54.78 | 49.37 | 50.69 | - | - | - | 56.00 | 54.80 |
| **ENEL-7B**[*] | **57.91** | **49.90** | **51.84** | 5.32 | 8.58 | 13.98 | **61.00** | **55.20** |

We perform $l$ iterations of geometry aggregation, resulting in $\{F_{\mathrm{agg}_i}^l\}_{i=1}^{M_l}$. To ensure that the LLM fully extracts the local information, we choose to perform further semantic modeling using $H$ LLM layers after aggregation operations. This allows the model to learn the interactions between local information while preventing the loss of fine-grained geometric details. Subsequently, from the $L$th layer, we perform $l$ iterations of geometry propagation. Following the grid unpooling strategy, we use the point-to-grid mappings to propagate the aggregated features $F_{\mathrm{agg}}^l$ from each grid cell back to its corresponding set of points, generating $\{F_{\mathrm{pro}_i}^1\}_{i=1}^{M_{l-1}}$. After $l$ iterations, we obtain point tokens of length $M$, which are then processed by the remaining LLM layers. After processing through H additional LLM layers, the geometry aggregation and propagation process is repeated.

**Experiments and Insights.** We conduct step-by-step experiments on the Hierarchical Geometry Aggregation strategy, sequentially evaluating the impacts of the number of aggregation and propagation operations ($l$), the number of LLM layers between aggregation and propagation ($H$), and the incorporation of the gated self-attention mechanism.

- **The best performance is achieved when $l$ is set to 3.** As shown in Table 4, performing three aggregation and propagation operations achieves 48.93% and 53.00% performance on captioning and classification tasks, respectively. Fewer aggregation layers limit the capture of local geometric information, while too many layers oversimplify spatial relationships. Setting $l = 3$ achieves balanced modeling of local and global structures and realizes sampling ratio of approximately 1/8.

- **Compared to setting $H$ to 4 or 8, the highest performance is achieved when $H$ is set to 2**. It reaches 54.25% and 49.56% on the classification and captioning tasks, respectively. The excessive number of LLM layers between aggregation and propagation can lead to the oversmoothing of the aggregated local information, resulting in the loss of local structural details.

- **The gated self-attention mechanism effectively improves performance**, reaching 55.55% and 51.03% on classification and captioning tasks, respectively. The adaptive control of attention output ensures that global contextual information is utilized only when necessary, preventing it from disrupting local geometric structures. Additionally, it allows the model to adjust to different tasks.

## 4 RESULTS AND VISUALIZATION

**Results.** In Table 5, on the Objaverse benchmark Deitke et al. (2023), ENEL-7B achieves a GPT score of 51.03% for 3D object captioning, setting a new SOTA. In traditional metrics, Sentence-BERT and SimCSE reach 48.79% and 49.52%, respectively, comparable to PointLLM-PiSA-13B. For 3D object classification, ENEL-7B outperformes prior encoder-based 3D LMMs with a GPT score of 55.55%. Given the same training dataset as PointLLM, these results validate the effectiveness of

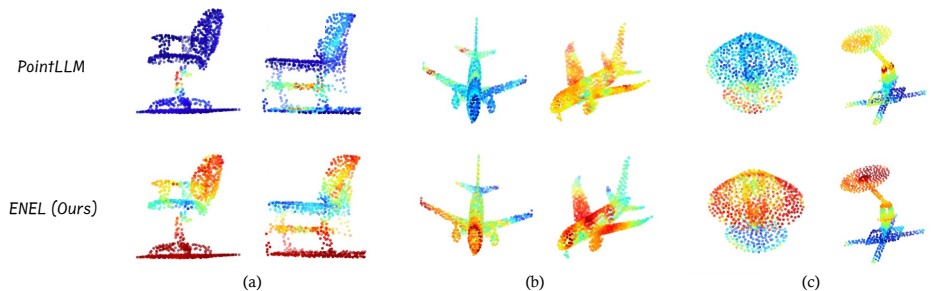

Figure 5: **Difference in Semantic Encoding.** By visualizing the attention scores of the average text token to the point tokens on the Objaverse dataset, we compare the semantic encoding potential of encoder-based and encoder-free architectures, where red indicates higher values. And (a) represents chairs, (b) represents airplanes, and (c) represents lamps.

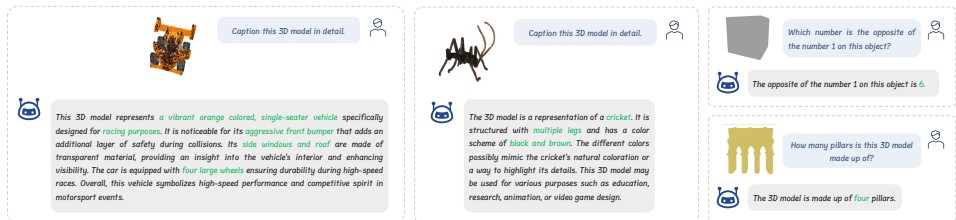

Figure 6: **ENEL Response Examples.** We demonstrate that ENEL is capable of providing accurate responses across different types of tasks, such as captioning and question answering, by effectively addressing a wide range of objects, including race cars, buildings, insects, and others.

our proposed **LLM-embedded Semantic Encoding** and **Hierarchical Geometry Aggregation** strategies for the encoder-free architecture. Additionally, on the 3D-VQA task of the 3D MM-Vet dataset Qi et al. (2024); Li et al. (2025b), despite the lack of spatial and embodied interaction-related data in the training set, ENEL achieves the GPT score of 43.8%, surpassing PointLLM-7B by 2.6%. Replacing 7B Vicuna with 13B Vicuna, ENEL-13B achieves substantial performance gains across tasks. When replacing the Vicuna-7B with Qwen2.5-7B and using ShapeLLM training data, ENEL-7B achieves over 6% improvements across benchmarks. Details of the evaluation metric and classification performance on the ModelNet dataset are provided in Appendix A.2.2 and A.2.3, respectively.

**Visualization.** In the Figure 5, we visualize the attention scores between the average text token and the point tokens in the last layer of both PointLLM and ENEL. Three object categories, including the chair, the airplane, and the desk lamp, are selected from the Objaverse dataset Deitke et al. (2023). In the Figure 5, red indicates higher values. We observe that in encoder-based 3D LMMs, the semantic relevance between the text tokens and the processed point tokens is relatively low. In contrast, ENEL, with its encoder-free architecture, achieves a high correlation between the features of the two different modalities, with the average text token focusing on key geometric structures of the objects, such as the backrest of the chair, the wings of the airplane, and the lampshade of the desk lamp.

**Response Visualization.** In the Figure 6, we present a visualization of ENEL's responses for both captioning and question answering (QA) formats. We observe that in the captioning task, ENEL can even accurately identify fine-grained categories such as a cricket. Moreover, in the QA task, ENEL effectively handles visual challenges such as general object recognition (e.g., reasoning about numbers on dice) and spatial reasoning (e.g., accurately interpreting building structures).

## 5 CONCLUSION

In this study, we investigate the potential of the encoder-free architecture in 3D understanding. Through a systematic analysis, we demonstrate that transferring the functionality of the 3D encoder to the LLM itself can effectively compensate for the performance degradation caused by the removal of the 3D encoder. To achieve this, we introduce the LLM-embedded Semantic Encoding strategy and the Hierarchical Geometry Aggregation strategy in the pre-training and instruction tuning stages. These strategies enable the encoding of high-level point cloud semantics while capturing critical local information. Our experiments highlight the promising prospects of the encoder-free architecture.

# 6 ACKNOWLEDGEMENT

This work is supported by Shanghai AI Laboratory.

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

# A APPENDIX

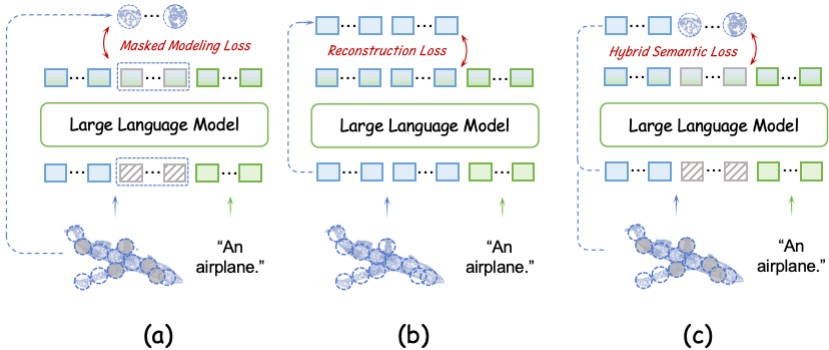

Figure 7: **Variants of Point Cloud Self-Supervised Learning Losses.** (a) The Variant of Masked Modeling Loss, (b) The Variant of Reconstruction Loss, (c) The Variant of Hybrid Semantic Loss.

## A.1 EXPERIMENTAL SETTINGS

**Implementation Details.** We use the LLaMA model Touvron et al. (2023) as our LLM backbone, with the 7B Vicuna-v1.1 Chiang et al. (2023) checkpoint as the default setting. In the token embedding layer, the point cloud is first processed by a linear layer to expand its dimension from 6 to 288. The input point cloud initially consists of 8192 points, followed by three iterations of farthest point sampling (FPS), reducing the size to 512, 256, and 128, respectively. After each FPS operation, k-Nearest Neighbors (k-NN) is applied with a cluster size of 81. And geometric features are extracted using triangular encoding, followed by linear layers that progressively increase the dimension to 576, 1152, and 2304. Finally, the projection layer maps the features to the LLM dimension of 4096. In the pre-training stage, we unfreeze the first four LLM layers. Within the LLM-embedded Semantic Encoding strategy, Hybrid Semantic Loss applies masked modeling to 30% of the tokens and reconstructs the patches for the remaining 70% visible tokens. During instruction tuning, geometric aggregation is applied at the end of the 1st, 2nd, and 3rd LLM layers to reduce point tokens. MaxMean pooling is used to retain more information. After two LLM layers, geometric propagation is applied at the end of the 6th, 7th, and 8th layers to restore the number of point cloud to 128. After two LLM layers, geometric aggregation is applied at the 11th–13th layers, followed by geometric propagation at the 16th–18th layers.

**Training and Evaluation Details.** During the two-stage training, each stage utilizes the same dataset and preprocessing method as PointLLM. All training are conducted on $4 \times 80G$ A100 GPUs in BF16 precision, utilizing FlashAttention, the AdamW optimizer, and a cosine learning rate schedule. During the pre-training stage, the model is trained for three epochs with a batch size of 128 and a learning rate of 4e-4. In the instruction tuning stage, it is conducted for three epochs with batch size of 32 and a learning rate of 2e-5. The GPT-4 model Achiam et al. (2023) used for classification and captioning tasks evaluation refers to "gpt-4-0613" version consistent with PointLLM Xu et al. (2023b). In contrast, the GPT-4 model employed for QA performance evaluation corresponds to "gpt-4-0125" version aligning with ShapeLLM Qi et al. (2024). Additionally, the GPT evaluation prompts for classification and captioning are identical to those used in PointLLM, while the prompts for QA follow those in ShapeLLM.

## A.2 MORE EXPERIMENTS

### A.2.1 VARIANTS OF POINT CLOUD SELF-SUPERVISED LEARNING LOSSES.

In the Figure 7, we exhibit the other variants of Masked Modeling Loss, Reconstruction Loss and Hybrid Semantic Loss.

As seen in Figure 7 (a), in the Masked Modeling Loss, after the learnable tokens are processed by the LLM, the tokens are transformed to the point patches $\{G_{\text{pre}_i}\}_{i=1}^{M*r} \in \mathbb{R}^{M*r \times k \times 3}$ through a linear layer. We utilize the $l_2$ chamfer distance to align the predicted $G_{\text{pre}}$ with the point patches $G_{mask}$

Table 6: **Ablation Experiments.** We begin the ablation experiments by changing the single configuration of the module from ENEL. $\Psi$ and $\Phi$ denote mask ratios of 60% and 30%, respectively. For Hybrid Semantic Loss, the subscripts $patch$ and $feat$ refer to the masked modeling target, with reconstruction targeting the corresponding $feat$ and $patch$. $l$ represents the number of aggregation and propagation operations. $H$ refers to the LLM layers between $l$ aggregation and $l$ propagation operations. $O$ refers to the LLM layer between two individual aggregation or propagation operations.

| Model | Cap | | | | | | Cls (Avg) |
|---|---|---|---|---|---|---|---|
| | GPT-4 | Sentence-BERT | SimCSE | BLEU-1 | ROUGE-L | METEOR | GPT-4 |
| **ENEL-7B** | 51.03 | 48.79 | 49.52 | 3.91 | 7.20 | 12.68 | 55.55 |
| –Hybrid Semantic Loss | 47.15 | 48.06 | 48.31 | 3.40 | 7.43 | 11.84 | 50.50 |
| Hybrid Semantic Loss$^{\Phi}_{patch}$ | 49.13 | 48.80 | 49.20 | 3.99 | 7.20 | 12.38 | 52.30 |
| Hybrid Semantic Loss$_{patch}{}^{\Psi}$ | 48.79 | 48.30 | 49.00 | 3.65 | 6.90 | 11.98 | 52.10 |
| Hybrid Semantic Loss$^{feat}$ $\Psi$ | 49.62 | 48.00 | 48.67 | 3.78 | 6.82 | 12.33 | 51.50 |
| –gate mechanism | 49.61 | 48.41 | 48.97 | 3.79 | 7.12 | 12.48 | 53.60 |
| l=2,H=2,O=0 | 48.83 | 48.20 | 48.53 | 3.72 | 6.89 | 12.01 | 51.50 |
| l=2,H=4,O=0 | 49.05 | 48.47 | 48.62 | 3.65 | 7.10 | 12.31 | 52.20 |
| l=2,H=2,O=2 | 48.96 | 47.95 | 48.88 | 3.80 | 7.05 | 12.55 | 52.00 |
| l=2,H=4,O=2 | 49.68 | 48.70 | 48.85 | 3.84 | 7.56 | 12.76 | 53.10 |

Table 7: Comparison of computational complexity between PointLLM-7B and ENEL-7B. S1 and S2 refer to the pre-training and instruction tuning stages, respectively. Conv. Steps indicates the number of steps required for loss convergence.

| Method | Time (H) | Memory (S1/S2) | FLOPs | Conv. Steps (S1/S2) |
|---|---|---|---|---|
| PointLLM-7B | 31.6 | 67G / 57G | $2.0 \times 10^{18}$ | 10100 / 4300 |
| ENEL-7B | 22.2 | 56G / 42G | $1.59 \times 10^{18}$ | 9790 / 3700 |
| **Improvement** | 29.7% | 16.4% / 26.3% | 20.5% | 2.9% / 14.0% |

corresponding to the masked tokens, reconstructing the spatial information. The optimization is:

$$\frac{1}{M*r}\sum_{i=1}^{M*r}\left(\min_{j}\|a_i - b_j\|_2^2 + \min_{j}\|b_i - a_j\|_2^2\right),\tag{9}$$

where $a = G_{\text{pre}}$ and $b = G_{mask}$.

As shown in Figure 7 (b), after the point feature tokens $\{F_i\}_{i=1}^M$ are encoded by the LLM, the Mean Squared Error (MSE) is computed between the predicted $\tilde{F}_{\text{pre}}$ and the ground truth $F$. The optimization can be written as

$$\mathcal{L}_{\text{mask}} = \frac{1}{M}\sum_{i=1}^{M}\left(\|F_{\text{pre}_i} - F_i\|_2^2\right).\tag{10}$$

Finally, in the Figure 7 (c) Hybrid Semantic Loss, the masked tokens and the corresponding patches are referred to as $\{F_{\text{mask}_i}\}_{i=1}^{M*r}$ and $\{G_{\text{mask}_i}\}_{i=1}^{M*r}$, respectively. The remaining tokens are denoted as $\{F_{\text{vis}_i}\}_{i=1}^{M*(1-r)}$ and $\{G_{\text{vis}_i}\}_{i=1}^{M*(1-r)}$. After passing point tokens through the LLM, we compute the MSE between $F_{\text{pre}}$ and $F_{\text{vis}}$. The learnable tokens $F_{\text{learn}}$ are transformed into $G_{\text{pred}}$, and the $L_2$ Chamfer distance is computed between $G_{\text{pred}}$ and $G_{\text{mask}}$. These two are added to the original cross-entropy loss with coefficients all equal to 1.

### A.2.2 METRIC ANALYSIS

**GPT-4 Evaluation** is a LLM-as-a-judge framework based on custom prompts. Given a model-generated description and human reference, GPT-4 identifies key attributes from the reference, measures how many are accurately or partially matched in the model output, and returns a score from 0 to 100 with a brief explanation. It offers more comprehensive and human-aligned evaluation.

**Traditional metrics** like BLEU measure n-gram precision, ROUGE-L uses longest common subsequence, and METEOR combines unigram precision and recall with lemmatization and synonym matching. However, these metrics struggle with semantic similarity and tend to favor shorter outputs.

Table 8: ModelNet40 classification results under instruction-typed and completion-typed prompts. The instruction-typed (I) prompt is "What is this?" and the completion-typed (C) prompt is "This is an object of."

| Model | ModelNet (I) | ModelNet (C) | ModelNet-Avg |
|---|---|---|---|
| PointLLM-7B | 53.44 | 51.82 | 52.63 |
| PointLLM-13B | 53.00 | 52.55 | 52.78 |
| ShapeLLM-7B | – | – | 53.08 |
| ShapeLLM-13B | – | – | 52.96 |
| PointLLM-PiSA-7B | 54.58 | 52.60 | 53.59 |
| PointLLM-PiSA-13B | 55.03 | 53.81 | 54.42 |
| ENEL-7B | 54.82 | 53.69 | 54.26 |
| ENEL-13B | 55.59 | 54.38 | 55.00 |
| **ENEL-7B*** | 61.25 | 60.47 | **60.86** |

**Reasons for low traditional metrics:** 3D-LLM with high traditional metric scores generates captions averaging 20 words—much shorter than ENEL and other methods. However, this does not indicate better output quality and performs worse in human evaluations. Traditional metrics often fail to assess the quality of detailed LLM outputs, as they favor shorter responses and struggle to capture semantic similarity. The GPT-4 score offers stronger semantic understanding, greater diversity, and better generalization.

*Examples:* Here is a typical example where GPT-4 gives high scores but traditional metrics give low scores. Given a point cloud of an airplane, the model outputs:

> *"The 3D model portrays a white cartoon airplane, styled in a simplistic and charming fashion. . . This model can be inferred to be used in animated children's media or as a playful element in a game or learning application design."*

The ground truth:

> *"This 3D object is an airplane with distinct wings and a tail. It has a long fuselage with glass windows at the front and sides. The round-shaped wings are located in the middle."*

The model correctly identifies the object as an airplane and captures key style features like simplicity, cartoon form, and whiteness. It also reasonably infers use in children's media, showing strong understanding. However, traditional metrics **rely on n-gram overlaps**. Phrases like "airplane body and wings" differ from the ground truth "fuselage with glass windows," leading to mismatches. The output is also longer and more descriptive, while the ground truth is concise and factual, and includes extra details like "white cartoon airplane," all contributing to low traditional scores.

### A.2.3 MODELNET CLASSIFICATION TASK

As shown in Table 8, ENEL-7B achieves an average accuracy of 54.26%, surpassing PointLLM-7B (52.63%), ShapeLLM-7B (53.08%) and PointLLM-PiSA-7B (53.59%). Similarly, ENEL-13B reaches 55.00%, outperforming both ShapeLLM-13B (52.96%) and PointLLM-PiSA-13B (54.42%). These results demonstrate the effectiveness of the encoder-free design in 3D object understanding.

### A.2.4 COMPLEXITY ANALYSIS

In Table 7, compared to PointLLM-7B, ENEL-7B demonstrates significant improvements while using the same training dataset. It achieves 29.7% faster training time, reduces GPU memory usage by 16.4% and 26.3% in Stage 1 and Stage 2, respectively, lowers training FLOPs by 20.5%, and accelerates convergence speed by 2.9% (Stage 1) and 14.0% (Stage 2).

### A.2.5 Encoder-Free Architecture Claim.

Following the consensus in recent Large Multimodal Model (LMM) literature, we strictly define an architecture as "encoder-free" based on two criteria: (1) the absence of a heavy, independently pretrained visual backbone, and (2) the utilization of end-to-end training from scratch. Unlike traditional 3D LMMs that rely on decoupled, pretrained encoders (e.g., Point-BERT Yu et al. (2022)) for semantic extraction, our design integrates a lightweight, randomly initialized embedding layer trained jointly with the LLM.

**Alignment with Community Standards.** This design philosophy parallels established encoder-free paradigms in the 2D image and video domains. For instance, EVE Diao et al. (2024a) utilizes a token embedding layer based on convolution and cross-attention (∼16M parameters), while ELVA Li et al. (2025a) employs a spatio-temporal attention layer (∼9M parameters) for video framing. Similarly, Mono-InternVL Luo et al. (2024) relies on a lightweight stack of convolutions (∼10M parameters). As detailed in Table 9, our proposed point embedding layer comprises only 3M parameters. This is not only significantly more lightweight than its 2D counterparts but also orders of magnitude smaller than typical 3D encoders (e.g., ∼88M for PointBERT used in PointLLM). Our module functions strictly as a *tokenizer* rather than a visual encoder.

Table 9: **Comparison of Tokenizer Parameters across Domains.**

| Method | Domain | Tokenizer Structure | Tokenizer Params | Ratio (Tok./Total) |
|---|---|---|---|---|
| EVE / EVEv2 Diao et al. (2024a; 2025) | Image | Conv + Cross-Attn | 16 M | ∼0.23% |
| Mono-InternVL Luo et al. (2024) | Image | Stacked Conv | 10 M | ∼0.14% |
| ELVA Li et al. (2025a) | Video | Spatio-temporal Attn | 9 M | ∼0.13% |
| PointLLM Xu et al. (2023a) | 3D | PointBERT Encoder | ∼88 M | ∼1.24% |
| **Ours** | **3D** | **Point Embedding** | **3 M** | **∼0.04%** |

**Structural Formatting vs. Semantic Encoding.** We explicitly distinguish the structural operations used in our embedding layer—specifically Farthest Point Sampling (FPS) and $k$-Nearest Neighbors ($k$-NN)—from semantic encoding. Due to the data irregularity of unstructured 3D point clouds, FPS and $k$-NN serve as the mathematically necessary equivalents of the "patchify" or "stride" operations used in 2D Vision Transformers. They are required to group raw data points into processable tokens. Crucially, these operations are *parameter-free*. The subsequent learnable MLPs serve only to project these local geometric groupings into the feature dimension required by the LLM.

### A.2.6 More Ablation Experiments

We begin the ablation experiments starting from the ENEL-7B, which is the reverse order compared to the experiments in the main text, as showcased in Table 6

**The Effects of LLM-embedded Semantic Encoding Strategy.** In the Table 6, on the basis of ENEL, removing the Hybrid Semantic Loss during the pre-training stage significantly degrades performance. The GPT-4 score for the captioning task drops from 51.03% to 47.15%, and the GPT-4 score for the classification task decreases to 50.50%. This is because the proposed self-supervised learning function for point clouds effectively captures the detailed structures and high-level semantics.

Based on ENEL-7B, we find that setting the mask ratio in the Hybrid Semantic Loss to 30% consistently yields better results than 60%. Additionally, the configuration where the masked token part predicts features while the visible token part reconstructs patches outperforms the reverse setting—where the masked token part predicts patches and the visible token part reconstructs features. This phenomenon can be explained as follows: a mask ratio of 30% retains critical information while facilitating the model to effectively utilize the visible tokens to derive the masked parts. When the mask ratio is set too high, the model fails to learn the global context knowledge adequately. Moreover, when the masked token part is tasked with predicting features, the model focuses on learning the high-level context semantics, while the patch reconstruction aids in accurately capturing low-level details. In contrast, when the masked token part predicts patches, the model becomes excessively dependent on local features during the process of semantic reconstruction.

**The Effects of Hierarchical Geometry Aggregation Strategy.** After removing the gating mechanism in the self-attention of the aggregation operation, the performance drops to 49.61% and 53.60% on the captioning and classification tasks, respectively. The gating mechanism helps the model

Table 11: **Generalization analysis on the ShapeLLM baseline.** $^\dagger$ denotes the model is implemented based on the ShapeLLM baseline.

| Model | Cap | | Cls | QA |
|---|---|---|---|---|
| | GPT-4 | S-BERT | GPT-4 | GPT-4 |
| PointLLM-7B | 44.85 | 47.47 | 53.00 | 41.20 |
| PointLLM-13B | 48.15 | 47.91 | 54.00 | 46.60 |
| ShapeLLM-7B | 46.92 | 48.20 | 54.50 | 47.40 |
| ShapeLLM-13B | 48.94 | 48.52 | 54.00 | 53.10 |
| ENEL-7B | 51.03 | 48.79 | 55.55 | 43.80 |
| ENEL-7B$^\dagger$ | 53.26 | 48.75 | 56.00 | 48.90 |
| ENEL-13B | 53.24 | 48.92 | 56.00 | 48.50 |
| ENEL-13B$^\dagger$ | **54.78** | **49.37** | **56.00** | **54.80** |

to adaptively filter information, allowing it to focus on more discriminative features. Without the dynamic adjustment to focus on different parts of the input, the generated text from the LLM lacks accuracy and coherence, leading to a decrease in performance.

As the number of aggregation and propagation operations decreases, overall performance declines, mainly due to insufficient layers failing to adequately model complex spatial relationships in point clouds. We observe that increasing the number of LLM layers between the final aggregation operation and the first propagation operation leads to improved performance. This suggests that fewer cascaded aggregation operations require deeper network architectures for high-level feature abstraction; otherwise, insufficient depth may lead to degraded hierarchical representations. Furthermore, the presence of LLM layers between each aggregation or propagation operation enhances performance by allowing the model to process and transform compressed information. Through self-attention mechanisms, these intermediate layers can recapture and restore details lost during the aggregation process.

**General Language Capabilities.** We investigate whether 3D instruction tuning compromises the LLM inherent text generalization. As observed in 2D LMMs (e.g., LLaVA), training exclusively on multimodal data often leads to catastrophic forgetting. As shown in Table 10, ENEL-7B achieves 46.4% on the MMLU benchmark Hendrycks et al. (2020), showing a slight decline compared to the original Vicuna-7B (47.1%). To isolate the cause, we first removed the *Hierarchical Geometry Aggregation* (HGA) strategy. The marginal difference (46.5% vs.

Table 10: Ablation on general language capabilities (MMLU).

| Method | MMLU (5-shot) |
|---|---|
| Vicuna-7B | 47.1 |
| ENEL-7B | 46.4 |
| *w/o* HGA | 46.5 |
| **+ Text Data** | **47.3** |

46.4%) indicates that the architectural design is not the primary factor in this degradation. However, by incorporating 12K pure text samples sampled from Evol-Instruct-GPT4-Turbo-143K Chen et al. (2024a) during the SFT stage, the performance rebounds to 47.3%, surpassing the original baseline. This confirms that mixing pure text data effectively mitigates catastrophic forgetting and preserves the general reasoning capabilities.

**Generalizability across Baselines.** To demonstrate that the effectiveness of our encoder-free architecture is not limited to a specific framework (i.e., PointLLM), we extend our evaluation to the ShapeLLM baseline. We replace the encoder-based design in ShapeLLM with our proposed ENEL architecture while strictly maintaining the original Vicuna-based LLM backbone, training data, and hyper-parameter settings for a fair comparison. As presented in Table 11, ENEL-7B$^\dagger$ achieves a GPT-4 score of 53.26% on the captioning task, significantly outperforming the original ShapeLLM-7B. Similar consistent improvements are observed at the 13B scale across diverse tasks.

Table 12: **Robustness to varying inference resolutions.** We evaluate models on the Objaverse Captioning task using GPT-4 scores across different point cloud densities (2K to 16K).

| Method | 2K/128 | 4K/256 | 8K/512 | 12K/1024 | 16K/2048 | Avg. |
|---|---|---|---|---|---|---|
| PointLLM-7B | 33.7 | 41.4 | 44.0 | 41.7 | 32.8 | 38.7 |
| ENEL-7B | 44.6 | 49.0 | 51.0 | 50.0 | 46.3 | 48.2 |
| ENEL-Mix-7B | **46.5** | **50.4** | **51.9** | **50.8** | **47.7** | **49.5** |

**Robustness to Resolution Variations.** A significant limitation of encoder-based 3D LMMs is their sensitivity to input resolution discrepancies between training and inference. To evaluate the adaptability of our encoder-free architecture, we introduce a variant named **ENEL-Mix**, which is pre-trained by randomly sampling point clouds ranging from 2K to 16K points per batch. We evaluate performance consistency across varying inference resolutions (2K–16K) on the Objaverse

Table 13: **Impact of mixed-resolution training on general tasks.** Comparing ENEL-Mix against standard baselines on Captioning, Classification, and QA tasks.

| Model | Cap | | Cls | QA |
|---|---|---|---|---|
| | GPT-4 | S-BERT | GPT-4 | GPT-4 |
| PointLLM-7B | 44.85 | 47.47 | 53.00 | 41.20 |
| ShapeLLM-7B | 46.92 | 48.20 | 54.50 | 47.40 |
| ENEL-7B | 51.03 | 48.79 | 55.55 | 43.80 |
| ENEL-Mix-7B | **51.92** | **49.06** | **56.00** | **44.40** |

Captioning task. As detailed in Table 12, the standard ENEL-7B already demonstrates superior robustness compared to PointLLM-7B, increasing the average GPT-4 score from 38.72 to 48.18 (+9.46) and significantly reducing performance variance. This indicates that the encoder-free design is inherently less brittle to density changes. Furthermore, ENEL-Mix-7B enhances stability, achieving the lowest relative performance drop (10.40%) between the best and worst resolutions. Beyond robustness, mixed-resolution training serves as an effective data augmentation strategy. As shown in Table 13, ENEL-Mix-7B yields consistent improvements across Captioning, Classification, and QA benchmarks, demonstrating that exposing the model to varying geometric densities facilitates a more generalized 3D understanding.

**Hyperparameter Sensitivity Analysis.** We evaluate the robustness to variations in loss weighting and auxiliary hyperparameters, with results summarized in Table 14. We perform a grid search for the auxiliary loss weights within the range $\{0.1, 0.5, 1.0, 2.0\}$ while fixing $\lambda_{LLM} = 1$. The model demonstrates high stability, with performance fluctuations restricted to within 1.0 point across the $[0.5, 2.0]$ interval. Extreme reductions (e.g., 0.1) lead to noticeable drops, validating the necessity of these semantic guidance components. Although $\lambda_{mask} = 0.5$ yields marginal gains, we adopt unit weighting ($\lambda = 1.0$) as the default for its simplicity and generalizability.

We further investigate the configuration of the alignment modules. For the contrastive objective, a low temperature ($\tau = 0.07$) proves critical for effective discriminative learning; increasing $\tau$ significantly degrades performance. Regarding Knowledge Distillation (KD), applying $L_2$ normalization to both student and teacher features prior to calculating the MSE loss results in better convergence compared to using raw feature magnitudes.

Table 14: **Hyperparameter Sensitivity Analysis.**

| Ablation Target | Setting | Cap | | | Cls |
|---|---|---|---|---|---|
| | | GPT-4 | S-BERT | SimCSE | GPT-4 |
| *Part I: Hybrid Semantic Loss Coefficients* | | | | | |
| Vary $\lambda_{mask}$ | $\lambda_{mask} = 0.1$ | 49.29 | 48.30 | 48.92 | 54.00 |
| | $\lambda_{mask} = 0.5$ | **51.64** | **48.92** | **49.70** | **56.00** |
| | $\lambda_{mask} = 1.0$ (Default) | 51.03 | 48.79 | 49.52 | 55.55 |
| | $\lambda_{mask} = 2.0$ | 50.71 | 48.55 | 49.17 | 55.50 |
| Vary $\lambda_{recon}$ | $\lambda_{recon} = 0.1$ | 49.87 | 48.12 | 48.85 | 54.00 |
| | $\lambda_{recon} = 0.5$ | 50.85 | 48.61 | 49.37 | 55.50 |
| | $\lambda_{recon} = 1.0$ (Default) | 51.03 | 48.79 | 49.52 | 55.55 |
| | $\lambda_{recon} = 2.0$ | 50.80 | 48.45 | 49.20 | 55.00 |
| *Part II: Contrastive & KD Hyperparameters* | | | | | |
| Contrastive ($\tau$) | $\tau = 0.07$ | 45.57 | 45.16 | 45.82 | 47.00 |
| | $\tau = 0.2$ | 45.23 | 44.92 | 45.29 | 46.00 |
| | $\tau = 0.5$ | 44.81 | 44.68 | 45.20 | 46.50 |
| KD (MSE) | w/o Normalization | 47.60 | 47.56 | 48.02 | 51.50 |
| | w/ Normalization | 47.98 | 48.10 | 48.63 | 52.50 |

## A.3 USE OF LLM

In this paper, the Large Language Model (LLM) is mainly used for grammar checking and spelling error detection.

A.4 MODEL OUTPUT

In Figure 8, we showcase more model output, where our ENEL provides precise and diverse responses with multi-modal 3D instruction input.

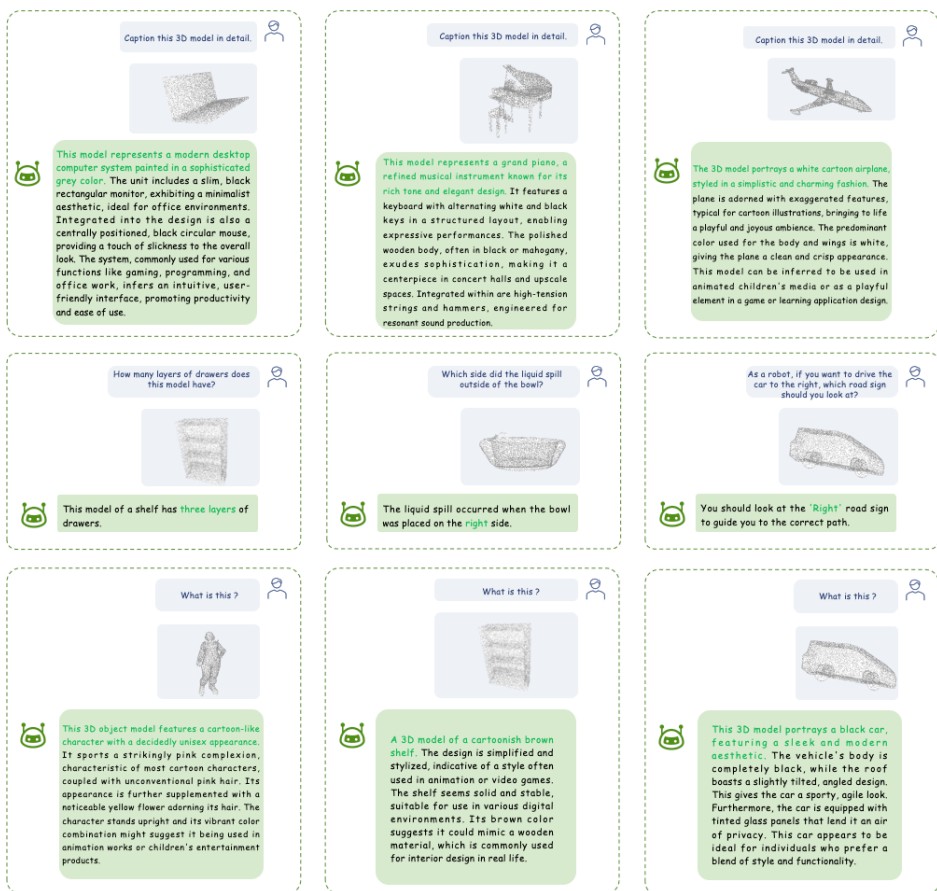

Figure 8: **ENEL Output Examples.** We demonstrate that ENEL provides precise and diverse responses when addressing different problems.

