# OpenReview forum: "Exploring the Potential of Encoder-free Architectures in 3D LMMs"
_ICLR.cc/2026/Conference — ICLR 2026 Poster_

### Official Review · Reviewer_fG4v · 2025-10-30

**Soundness:** 2
**Presentation:** 3
**Contribution:** 2
**Rating:** 4
**Confidence:** 5

**Summary:**

This paper introduces ENEL, an "encoder-free" 3D LMM. It features a pretraining scheme with a Hybrid Semantic Loss for learnable early LLM layers and a Hierarchical Geometry Aggregation strategy (gridding, attention, pooling/unpooling) for instruction tuning. The authors report their 7B model matches or exceeds encoder-based baselines on Objaverse captioning, classification, and VQA, supported by detailed ablations.

**Strengths:**

This paper poses the ambitious question of whether 3D LMMs can be "encoder-free." It clearly articulates two key challenges motivating this approach—resolution mismatch and semantic misalignment—using qualitative visualisations. To address these, the authors present a concrete system (ENEL) featuring two main contributions:

1.  A thorough empirical exploration of self-supervised objectives (including masked modelling, reconstruction, KD, and contrastive), which culminates in a final "Hybrid" loss for pretraining.
2.  The authors report competitive results against strong baselines like PointLLM and PointLLM-PiSA, while also noting potential compute and memory savings compared to PointLLM.

**Weaknesses:**

1.	“Encoder-free” claim is overstated.
The “token embedding module” is explicitly a lightweight Point-PN–style hierarchy: FPS downsampling, k-NN grouping, and learnable layers repeated 2–4 times, followed by a projection. This is a parameter-efficient encoder, not the near-identity/VQ/linear tokenizer style commonly implied by “encoder-free” in 2D works. The paper itself calls it “a lightweight variant of Point-PN.” Please either narrow the claim or compare against true encoder-free tokenizers (e.g., single linear/conv projection without FPS/k-NN).

	2.	Hierarchy moved into the LLM rather than removed.
The Hierarchical Geometry Aggregation repeatedly performs grid partitioning, intra-cell self-attention with a gate, mean-pooling, then later propagation (unpooling). Functionally, this recreates an encoder-like local-to-global hierarchy inside the decoder stack; it’s more of an architectural relocation of parameters than removal of hierarchical encoding. A head-to-head compute/param attribution (how many FLOPs/params sit in token-embedding + aggregation blocks vs a standard 3D encoder) is needed to justify “encoder-free” beyond naming.

	3.	Loss weighting fixed to 1 without principled tuning.
The Hybrid Semantic Loss sums masked-token prediction and patch reconstruction with coefficients all set to 1, then adds the language CE loss—again with unit weight. This choice is arbitrary; one would expect a meaningful sensitivity analysis or a schedule (e.g., ramping reconstruction, temperature/ratio sweeps). Currently, the paper states “coefficients all equal to 1” with no further study.

	4.	Fairness of comparisons / settings.
Several results swap backbones or data (e.g., ENEL-7B* uses Qwen2.5-7B and ShapeLLM data), while SOTA comparisons use different sizes. Stronger controls are needed: same LLM, same data, same token budget, and matched FLOPs/epoch, to isolate the architectural effect.

	5.	What is the minimal “tokenizer” needed?
Table 1 suggests deeper token-embedding helps (3 layers best), but the “-Encoder” baseline collapses. This invites a systematic study from pure linear projection → 1-layer MLP → Point-PN-lite (2–4 layers) at matched token counts, to identify the true necessity of FPS/k-NN (and their cost).

	6.	Compute story remains ambiguous.
Table 8 claims wall-clock/VRAM/FLOP gains vs PointLLM, but the new costs introduced by repeated grid builds, gated self-attention within many cells, and propagation are not separately profiled; neither is the token-embedding block. A per-stage FLOP/param breakdown would clarify where savings come from.

	7.	Evaluation choices.
Heavy reliance on GPT-4 scores for captioning is common, but adding human preference or task-specific structured metrics (beyond Sentence-BERT/SimCSE) would strengthen claims—especially since the method tunes the text generator itself. (No citation needed; suggestion.)

**Questions:**

1. “Encoder-free” claim is not substantiated. The proposed pipeline still performs hierarchical local aggregation (FPS/k-NN grouping, grid partitioning, intra-cell/gated self-attention, pooling/unpooling). Functionally, this is an encoder—even if its parameters are “moved” into early LLM blocks. Unless “encoder-free” is defined merely as “no separate backbone module,” the claim is misleading.

2. Sweep the Hybrid loss coefficients (e.g., λmask, λrecon ∈ {0.1, 0.3, 1, 3}) and report stability/performance; try temperature/normalization changes for contrastive/KD, or curriculum schedules.

3. Minimal tokenizer & hierarchy:
(a) Replace FPS+k-NN with a single linear projection (no sampling) to bound the true need for geometric preprocessing.
(b) Compare your aggregation to simple param-efficient adaptations closer to 2D “encoder-free” (e.g., LoRA on early LLM blocks; decoupled FFN) at equal parameter/FLOP budgets.

4. You motivate with resolution mismatch (Fig. 1). Please add a systematic resolution sweep for all tasks with matched token budgets, and report failure modes.

5. Provide a per-module FLOP/param/time breakdown for PointLLM vs ENEL: token embedding, aggregation/propagation (including grid building), and LLM compute.

---

> ### Author Response · Authors · 2025-11-24
>
> > **W1,Q1:Clarification on the “Encoder-free” Claim.**
>
> We appreciate the reviewer's examination of the “encoder-free” definition. In section 3.1 of main text, following the consensus in recent LMM literature, we strictly define “encoder-free” based on two criteria: (1) No heavy, independently pretrained visual backbone, and (2) End-to-end training from scratch. Below, we substantiate why ENEL falls squarely into this category and differs fundamentally from encoder-based methods.
>
> **1. Alignment with Community Standards (2D/Video LMMs).**
> The term “encoder-free” in the LMM community typically refers to removing the pretrained vision encoder (e.g., CLIP-ViT) in favor of a lightweight, randomly initialized embedding layer (tokenzier) trained jointly with the LLM.
> In 3D domain, pretrained encoders are characterized by two key features: (1) independent pretraining on point cloud tasks (e.g., reconstruction); (2) structural decoupling via projection layers connecting to the language model.
> Our design parallels accepted encoder-free works in 2D/Video domains:
>
> * **EVE/EVEv2 [NeurIPS'24/ICCV'25]:** Uses a token embedding layer (16M params) with convolution and cross-attention.
> * **ELVA [ICCV'25]:** A Video-LLM with a 9M-parameter embedding layer involving attention for long-range spatio-temporal framing.
> * **Mono-InternVL [CVPR'25]:** Uses a 10M-parameter stack of convolutions and linear layers.
>
> Compared to these methods, our point embedding layer is even more lightweight (only 3M parameters), significantly smaller than the typical 3D encoder (e.g., PointBERT is ~88M).
> Our point embedding layer is not independently pretrained but jointly trained from scratch with other ENEL components.
> It functions as a *tokenizer* rather than a visual encoder.
> | Method | Domain | Tokenizer Structure | Tokenizer Params | Param Ratio (Tokenizer/Total) |
> | :--- | :--- | :--- | :--- | :--- |
> | EVE / EVEv2[2,3] | Image | Conv + Cross-Attn | 16 M | \~0.23% |
> | Mono-InternVL[4] | Image | Stacked Conv | 10 M | \~0.14% |
> | ELVA[1] | Video | Spatio-temporal Attn | 9 M | \~0.13% |
> | PointLLM | 3D | PointBERT Encoder | \~88 M | \~1.24% |
> | **Ours** | **3D** | **Point Embedding** | **3 M** | **\~0.04%** |
>
> **2. Why FPS/k-NN in embedding layer?** We clarify that in 3D domain, these are structural formatting operations, not semantic encoding operations.
>
> * **Data Irregularity:** Unlike 2D images defined on regular grids, 3D point clouds are unstructured. FPS and k-NN are the mathematically necessary equivalents of "patchify" or "stride" for 3D data to group points into processable tokens.
> * **Parameter-Free:** FPS and k-NN operations are parameter-free. The learnable MLPs merely project these geometric groupings into the dimension required by the LLM.
> * **Geometry-Centric Tokenization:** Pretrained encoder extracts high-level semantics before the LLM sees it. Our embedding layer only aggregates local geometry. The semantic interpretation happens entirely within the LLM layers, aided by our proposed Hybrid Semantic Loss and Hierarchical Geometry Aggregation Strategy.
>
> **3. Why Hierarchical Geometry Aggregation Does Not Break Encoder-free Definition?**
> To explore the potential of encoder-free architectures in 3D LMMs, we focus on effectively transferring the functionality of 3D encoders into the LLM.
> Our Hierarchical Geometry Aggregation Strategy is designed to **inject 3D inductive bias into the LLM in a lightweight yet effective manner**, similar to how encoder-free 2D/Video LMMs incorporate inductive-bias mechanisms such as ELVA's bottom-up hierarchical design that progressively merges video tokens layer by layer and convolutional spatial bias in EVE/EVEv2.
> These inductive-bias mechanisms are not tied to the definition of an “encoder”, but simply enhance geometric awareness.
> Moreover, most of the 3D pretrained encoders such as *ReCon*, *ACT*, and *Uni3D* **do not rely on local-to-global hierarchy**.
>
> [1] Breaking the Encoder Barrier for Seamless Video-Language Understanding. ICCV 2025
>
> [2] Unveiling Encoder-Free Vision-Language Models. NeurIPS 2024 Spotlight
>
> [3] EVEv2: Improved Baselines for Encoder-Free Vision-Language Models. ICCV 2025 Highlight
>
> [4] Mono-InternVL: Pushing the Boundaries of Monolithic Multimodal Large Language Models with Endogenous Visual Pre-training. CVPR 2025

---

> > ### Comment · Reviewer_fG4v · 2025-11-28
> >
> > I thank the authors for their detailed response and addressed most of my concerns. However, I still have concerns regarding the architectural comparison.
> >
> > Specifically, methods like EVEv2, Mono-InternVL, and ELVA do not add additional attention layers to the LLM; instead, they decouple the model into vision and text components, tuning the vision weights (which are initialized using the pre-trained LLM weights). In contrast, the proposed method appears to (can be treated as ) move the gated attention layer from the vision component into the LLM backbone.
> >
> > Given this distinction, I am very interested in the performance impact of this design choice. Could the authors provide results where the gated attention layer is placed within the vision part rather than the LLM?

---

> ### Author Response · Authors · 2025-11-24
>
> > **W5,Q3(a): Experiments of Tokenzier.**
>
> We evaluate a progressive series of tokenizers based on the PointLLM with its original encoder removed: (1) pure projection without designed sampling (**A0**/**A1**), (2) FPS+k-NN preprocess with simple projectors (**B0**/**B1**/**B2**), and (3) Point Embedding with 2--4 layers (**C0**/**C1**/**C2**). All variants are evaluated under *matched token counts*, *identical token dimensions*, and the *same model architecture* to isolate the effect of tokenizer design.
> The results in table reveal a clear and consistent trend:
>
> **(i) Pure projection baselines perform the worst.**
> We use random sampling for token downsampling.
> The linear projector **A0** achieves only 26.98 and 30.00 GPT-4 scores on Captioning and Classification tasks, while adding non-linearity in **A1** yields marginal improvements to 27.49 and 30.25. This demonstrates that a learnable projector alone is insufficient for 3D point clouds.
>
> **(ii) Adding FPS+k-NN geometric preprocessing yields substantial gains.**
> We replace random sampling with FPS+k-NN for downsampling.
> Compared to **A0**, **B0** improves performance on Captioning and Classification tasks to 29.34 and 31.50, while **B1** further reaches 30.66 on Captioning. The Mini-PointNet baseline **B2** (“-Encoder” setting) achieves GPT-4 scores of 33.37 and 35.50. These results strongly confirm the necessity and benefit of FPS+k-NN preprocessing for effective geometric grouping.
> However, the fact that patch-only baselines still fall significantly short of hierarchical architectures indicates the need for further aggregation.
>
> **(iii) Point Embedding provides the dominant improvements.** With only 2 layers, **C0** achieves Captioning and Classification performance of 38.85 and 40.60 with just 0.833M parameters, surpassing all simpler baselines. Performance peaks at 3 layers (**C2**), reaching 41.36 and 45.55 with 3.490M parameters, and saturates at 4 layers. This trend precisely matches our original findings and confirms that *hierarchical geometric processing*, rather than mere depth or parameter count, drives the observed improvements.
>
> | Tokenizer | Params (M) | Cap | | | Cls |
> |-------|-----|-----|-----|---|--|
> | | | **GPT-4** | **Sentence-BERT** | **SimCSE** | **GPT-4** |
> | A0: Linear | 0.887 | 26.98 | 36.95 | 38.12 | 30.00 |
> | A1: 1-MLP | 1.551 | 27.49 | 37.28 | 38.05 | 30.25 |
> | B0: FPS+kNN + Linear | 0.887 | 29.34 | 39.06 | 39.71 | 31.50 |
> | B1: FPS+kNN + 1-MLP | 1.551 | 30.66 | 39.42 | 40.10 | 32.00 |
> | B2: Mini-PointNet (original tokenizer ) | 4.818 | 33.37 | 41.19 | 41.68 | 35.50 |
> | C0: Point Embedding (2 layers) | 0.833 | 38.85 | 43.25 | 44.16 | 40.60 |
> | C1: Point Embedding (3 layers)$^\dagger$ | 3.490 | 41.36 | 44.82 | 45.59 | 45.55 |
> | C2: Point Embedding (4 layers) | 8.800 | 40.47 | 43.50 | 43.91 | 43.00 |
> *Note: $^\dagger$ Denotes our proposed default setting (Ours).*

---

> ### Author Response · Authors · 2025-11-24
>
> > **W2,Q3(b): Efficiency and Necessity of Hierarchical Geometry Aggregation Strategy.**
>
> Our Hierarchical Geometry Aggregation (HGA) Strategy is designed to **inject 3D spatial inductive bias into the LLM in a lightweight and efficient manner**, akin to inductive bias mechanisms in encoder-free 2D/Video LMMs (e.g., ELVA's bottom-up token merging and EVE's convolutional bias), **rather than merely relocating encoder parameters**. We demonstrate this via quantitative comparisons against both standard 3D encoders and simple 2D adaptations.
>
> **1. Quantitative Comparison.**
> We compare against two baselines under strict budget constraints: (1) **Standard 3D Encoder** which uses a pre-trained 3D encoder (like Point-BERT) coupled with the LLM, and (2) **Simple 2D Adapter** where, to avoid the excessive parameter and computational overheads of decoupled FFNs, we opt to apply LoRA to all linear layers within the first six LLM layers under the encoder-free settings. As shown in the table below, we report the parameters of the additional modules (excluding the LLM base) and the corresponding total training FLOPs associated with these extra modules. Performance refers to the GPT-4 score on the Objaverse Captioning & Classification task.
>
> | Setup | # Params (M) | FLOPs | Performance |
> | :--- | :---: | :---: | :---: |
> | **Standard 3D Encoder** | 88 | $1.64 \times 10^{17}$ | 44.85/53.00 |
> | **Simple 2D Adapter** | 30 | $1.51 \times 10^{17}$ | 48.87/53.00 |
> | **Ours (Tokenizer + HGA)** | 29 | $4.10 \times 10^{16}$ | 51.03/55.55 |
>
> **2. Hierarchical Geometry Aggregation Strategy is Lightweight, Not a “Relocated” Encoder.**
> The comparison above highlights that the Hierarchical Geometry Aggregation Strategy is fundamentally different from a standard 3D encoder, characterized specifically by a **Massive Reduction in Compute**. While a standard 3D encoder typically requires heavy stacks of self-attention layers, our strategy introduces only a fraction of the FLOPs and parameters, reducing total FLOPs by **70%+** compared to the encoder-based baseline. This confirms that Hierarchical Geometry Aggregation Strategy is not a “relocation” of a heavy pretrained encoder but a highly optimized *compression* module designed to inject 3D spatial inductive bias into the LLM.
>
> **3. Necessity of Geometry Aggregation.**
> Comparing our method with Simple 2D Adapter justifies the existence of the Hierarchical Geometry Aggregation Strategy.
> Despite allocating an equivalent parameter budget to LoRA across the first six LLM layers, its involvement in both training stages results in FLOPs comparable to a standard 3D encoder, accompanied by a significant performance drop. This indicates that point cloud data possesses complex geometric structures that simple linear projections cannot effectively capture. In contrast, the Hierarchical Geometry Aggregation Strategy avoids the heavy computational burden of the 3D encoder while providing the essential geometric abstraction that 2D adapters lack.
>
> -----
>
> > **W6,Q5: Detailed Computational Breakdown.**
>
> We provide a granular breakdown of training FLOPs, distinguishing between **Visual Processing Modules** (A & B) and the **LLM Backbone** (C). This analysis confirms that the primary computational burden in the baseline stems from the $O(N^2)$ global self-attention of the standard encoder, whereas our efficiency gains derive from two key optimizations:
>
>   * **Negligible Hierarchical Geometry Aggregation Overhead:** Grid partitioning and propagation are primarily non-parametric operations with **$O(N)$** complexity. By isolating these costs, we demonstrate that the strategy consumes only $\mathbf{0.11 \times 10^{17}}$ FLOPs—an order of magnitude lower than the baseline's visual cost and negligible compared to the LLM.
>
>   * **Source of Savings:** The total **20.5%** FLOPs gain is driven by:
>     1.  **Visual Stage (A+B):** Replacing the heavy Point-BERT with our lightweight tokenizer and Hierarchical Geometry Aggregation Strategy achieves a **4x efficiency boost** in the visual domain.
>     2.  **LLM Optimization (C):** The effective token aggregation (via pooling) reduces sequence lengths, thereby alleviating the computational load on subsequent layers and saving an additional $\mathbf{2.8 \times 10^{17}}$ FLOPs within the LLM backbone itself.
>
> | Module Component | PointLLM-7B|  | ENEL-7B| |
> | :--- | :---: | :---: | :---: | :---: |
> |  | **# Params(M)**|**Training FLOPs** | **# Params(M)**|**Training FLOPs** |
> | **A. Visual Stage / Tokenizer** | 88.0 | $1.64 \times 10^{17}$ | 3.5 | $0.30 \times 10^{17}$ |
> | **B. Adapter / HGA** | 20.0 | $1.35 \times 10^{17}$ | 25.2 | $0.11 \times 10^{17}$ |
> | **C. LLM Backbone (Vicuna-7B)** | 7000 | $18.3 \times 10^{17}$ | 7000 | $15.5 \times 10^{17}$ |
> | **Total Model** | 7.1 B | $2.00 \times 10^{18}$ | 7.0 B | $1.59 \times 10^{18}$ |

---

> ### Author Response · Authors · 2025-11-24
>
> > **W3,Q2: Hyperparameter Sensitivity and Loss Weighting.**
>
> We appreciate the valuable suggestion. To address this, based on ENEL-7B, we conduct a comprehensive **sensitivity analysis and ablation study** covering both loss coefficients and temperature settings.
>
> **1. Sensitivity Analysis of Loss Coefficients.**
> We formulate the total loss as $L _ {total} = \lambda _ {LLM} \mathcal{L} _ {CE} + \lambda _ {mask} \mathcal{L} _ {mask} + \lambda _ {recon} \mathcal{L} _ {recon}$. We performe a grid search for $\lambda_{mask}, \lambda_{recon} \in$ {0.1, 0.5, 1.0, 2.0} while keeping $\lambda_{LLM}=1$.
> As shown in the following table, we observe that:
> * **Stability:** The model performance is relatively robust to variations in both $\lambda_{mask}$ and $\lambda_{recon}$ within the range of $[0.5, 2.0]$.
>     Specifically, the performance fluctuations on the primary metrics remain **within 1.0 point**.
>     Extreme values (e.g., $0.1$) lead to noticeable performance drops, validating the necessity of the semantic guidance.
> * **Optimal Setting:** While setting $\lambda_{mask}=0.5$ yields a marginal gain (51.64 vs. 51.03 on Captioning), the default setting ($\lambda=1.0$) remains highly competitive. It indicates that unit weighting is an effective and generalizable default choice.
>
> | Ablation Target |  Setting | Cap | |  | Cls |
> | :--- | :--- | :--- | :--- | :--- | :--- |
> | | |GPT-4 |S-BERT | SimCSE | GPT-4 |
> | |**Part I: Hybrid Semantic Loss Coefficients** | | | | |
> | Vary $\lambda_{mask}$ | $\lambda_{mask}=0.1$ | 49.29 | 48.30 | 48.92 | 54.00 |
> | | **$\lambda_{mask}=0.5$** | **51.64** | **48.92** | **49.70** | **56.00** |
> | | $\lambda_{mask}=1.0$ (Baseline) | 51.03 | 48.79 | 49.52 | 55.55 |
> | | $\lambda_{mask}=2.0$ | 50.71 | 48.55 | 49.17 | 55.50 |
> | Vary $\lambda_{recon}$ | $\lambda_{recon}=0.1$ | 49.87 | 48.12 | 48.85 | 54.00 |
> | | $\lambda_{recon}=0.5$ | 50.85 | 48.61 | 49.37 | 55.50 |
> | | $\lambda_{recon}=1.0$ (Baseline) | 51.03 | 48.79 | 49.52 | 55.55 |
> | | $\lambda_{recon}=2.0$ | 50.80 | 48.45 | 49.20 | 55.00 |
> | |**Part II: Contrastive & KD Hyperparameters** | | | | |
> | Contrastive ($\tau$) | $\tau = 0.07$ (Baseline) | 45.57 | 45.16 | 45.82 | 47.00 |
> | | $\tau = 0.2$ | 45.23 | 44.92 | 45.29 | 46.00 |
> | | $\tau = 0.5$ | 44.81 | 44.68 | 45.20 | 46.50 |
> | KD (MSE) | w/o Normalization (Baseline) | 47.60 | 47.56 | 48.02 | 51.50 |
> | | w/ Normalization | 47.98 | 48.10 | 48.63 | 52.50 |
>
> **2. Ablation on Contrastive/KD Temperature & Normalization.**
> Based on ENEL-7B, we also investigate the temperature parameter $\tau$ and normalization strategies for the Contrastive/KD component.
> * **Contrastive Temperature ($\tau$):** In the contrastive learning objective, $\tau$ controls the smoothness of the distribution. We swept $\tau \in$ {0.07, 0.2, 0.5}. Results show that a lower temperature ($\tau=0.07$) is critical for effective contrastive learning.
> * **KD Normalization:** We investigate Feature Normalization on KD loss. We find that applying $L_2$ normalization to both student and teacher features before MSE improves convergence compared to raw features.
>  ---
> > **W4: Fairness of Comparisons.**
>
> Sorry for the confusion. We emphasize that the standard ENEL-7B/13B models are **already strictly aligned with the PointLLM baseline**, utilizing the identical Vicuna v1.1 LLM backbone, training data and training epoches. Under this controlled configuration, ENEL-7B significantly outperforms PointLLM-7B (51.03 vs. 44.85 on Captioning task), confirming that the performance improvements are **attributed to our architectural design** rather than disparities in model size or data.
> However, the ENEL-7B* variant is included to demonstrate the scalability of our method when equipped with a stronger LLM backbone, rather than a direct comparison to older baselines.
>
> To further address concerns regarding varying experimental settings and to demonstrate the robustness of our architecture, we conduct an additional experiment strictly aligned with the ShapeLLM configuration. By adopting the same LLM backbone and utilizing the ShapeLLM training data, we ensure a direct comparison. As shown in the table below, our method clearly outperforms ShapeLLM under these identical constraints, further validating the efficacy of our design.
>
> | Model | Cap |  | Cls |QA |
> |-------|-----|-----|-----|---|
> |  | **GPT-4** | **Sentence-BERT** | **GPT-4** |**GPT-4** |
> | ShapeLLM-7B | 46.92 | 48.20 | 54.50 | 47.40 |
> | ShapeLLM-13B | 48.94 | 48.52 | 54.00 | 53.10 |
> | ENEL-7B$^{\dagger}$ | 53.26 | 48.75 | 56.00 | 48.90 |
> | ENEL-13B$^{\dagger}$ | 54.78 | 49.37 | 56.00 | 54.80 |
>
> Note: † indicates models built on ShapeLLM baseline.

---

> ### Author Response · Authors · 2025-11-24
>
> > **W7: Evaluation Choices.**
>
> We appreciate the suggestion to incorporate human preference. To ensure a rigorous and standardized comparison, we adopt the human evaluation protocol established by PointLLM.
> We conduct a **blind review** on objects from Objaverse Captioning task via the official *Objaverse Explorer*. We evaluate performance based on the **Correctness Score**, which awards points for each distinct correct attribute (e.g., category, color, shape), with partial credit ($0\text{-}1$) for semi-accurate descriptions.
>
> We performe a pairwise comparison between **ENEL** and **PointLLM**. Based on the *Correctness Score*, we calculate the Win Rate, as defined in PointLLM. As shown in the table, ENEL achieves a **Win Rate of 67%** against PointLLM. These results corroborate our automated metrics (GPT-4/SimCSE), confirming that ENEL generates captions that are semantically richer and more faithful to the 3D visual content.
>
> | Pairwise Comparison | Metric | **Win** | **Tie** | **Lose** |
> | :--- | :--- | :---: | :---: | :---: |
> | ENEL-7B vs. PointLLM-7B | Human Eval | **67%** | 12% | 21% |
> ---
>
> > **Q4: Resolution Sweep and Failure Modes.**
>
> We performe a systematic resolution sweep across all three tasks (Captioning, Classification, and QA) on the Objaverse dataset to validate our motivation. We evaluate input point cloud resolutions ranging from 2K to 16K. Crucially, to ensure a fair comparison, we maintain a matched token budget by adjusting the pooling ratio accordingly. Furthermore, we train ENEL-Mix-7B by randomly sampling 2K-16K points per batch during pre-training, which is better suited for encoder-free architectures. As shown in the table below, the results demonstrate that the encoder-free ENEL-7B consistently outperforms baselines, while mixed-resolution training further enhances resolution robustness. The encoder-free architectures effectively achieve the better trade-off.
>
> | Resolutions | Cap|  |  | Cls | |  | QA |  | |
> | :--- | :--- | :--- | :--- | :--- | :--- | :--- | :--- | :--- | :--- |
> | | PointLLM-7B| ENEL-7B | ENEL-Mix-7B | PointLLM-7B| ENEL-7B | ENEL-Mix-7B| PointLLM-7B| ENEL-7B | ENEL-Mix-7B|
> | 2K | 35.3 | 45.1 | **46.9** | 46.25 | 52.00 | **53.75** | 32.10 | 39.80 | **41.30** |
> | 4K | 42.1 | 49.5 | **50.6** | 51.00 | 55.00 | **55.50** | 39.30 | 42.50 | **43.60** |
> | 8K | 44.0 | 51.0 | **51.9** | 53.00 | 55.55 | **56.00** | 41.20 | 43.80 | **44.40** |
> | 12K | 41.6 | 50.0 | **50.7** | 51.25 | 55.50 | **56.00** | 39.00 | 42.90 | **43.50** |
> | 16K | 32.5 | 46.2 | **47.7** | 45.50 | 51.75 | **54.00** | 33.60 | 40.00 | **41.20** |
>
> We conduct a qualitative analysis to identify specific failure modes at extreme resolutions, visualization of which has been provided in the Figure 8 of the revised main text PDF.
> For **low-resolution failure**, the primary failure mode is *geometric aliasing*. Fine-grained structural details (e.g., the thin handle of a mug or legs of a chair) are lost during discretization, leading to misclassifications or generic captions (e.g., labeling a “mug” as a “cylinder”).
> For **high-resolution failure**, under a fixed token budget, high-resolution inputs force the aggregation of spatially distant features, leading to *feature dilution*. In captioning tasks, this often results in “hallucinations,” where the model describes textures or details that do not exist, likely due to noise sensitivity in the dense grid.

---

> ### Author Response · Authors · 2025-11-25
> **Summary of PDF Revisions**
>
> We sincerely appreciate the reviewer’s insightful feedback and remain committed to strengthening the paper. The following key revisions have been incorporated:
>
> * **Appendix A.3.5:** Formalized the definition of **"Encoder-free"** architecture and provided a comparative analysis against established 2D/Video LMM standards.
> * **Appendix A.3.6:** Added comprehensive **Ablation Studies** covering:
>     * Tokenizer Architecture.
>     * Hyperparameter Sensitivity Analysis.
>     * Generalizability across Baselines.
> * **Appendix A.3.7:** Presented **resolution sweep results** across different tasks, along with the **visualization** of specific failure modes.
>
> ---
> These updates are highlighted in **blue** for easy reference.

---

> ### Author Response · Authors · 2025-11-27
> **Conclusion of Rebuttal and Invitation for Re-evaluation**
>
> Dear Reviewer fG4v,
>
> We remain highly appreciative of your time, effort, and insightful critiques on our manuscript.
>
> We have incorporated comprehensive new experiments and detailed analyses, which significantly strengthen the core claims of our paper. As the rebuttal phase concludes, we kindly invite your attention to these updates to confirm that all raised concerns have been comprehensively addressed.
>
> We hope that these improvements warrant a positive re-evaluation of our work. Should any residual issues remain, we are prepared to provide any further comprehensive experiments.
>
> Best regards,
>
> Paper 23290 Authors

---

> ### Author Response · Authors · 2025-11-28
> **Follow-up on Rebuttal: Discussion Phase Closing Soon**
>
> **Dear Reviewer fG4v**,
>
> We sincerely appreciate your valuable comments, which have greatly contributed to improving our work.
>
> Following your suggestions, we provide **a rigorous definition of encoder-free in Appendix A3.5 and compare it against accepted 2D/Video encoder-free LMMs** to validate our formulation. In Appendix A3.6, we **verify the necessity of our tokenizer architecture, conduct an ablation study on the sensitivity to loss weighting coefficients, and present a comparison of parameter counts and FLOPs across different components**. Finally, Appendix A3.7 includes **resolution sweep results across various tasks**, along with visualizations of representative failure modes.
>
> We hope that these revisions and clarifications address your concerns. If you find our responses satisfactory, we would greatly appreciate your reconsideration of our score. Otherwise, please feel free to let us know if you have any additional questions—we would be happy to provide further clarification.
>
> Thank you again for your insightful comments, which have helped us enhance the clarity and completeness of our work.
>
> Sincerely,
>
> All authors of Submission 23290

---

> ### Author Response · Authors · 2025-11-28
> **Response regarding Architectural Comparisons**
>
> We thank the reviewer for the insightful comment. In the realm of 2D/Video encoder-free LMMs, architectural strategies vary significantly to address specific domain challenges. Models like **EVEv2** and **Mono-InternVL** decouple visual processing by duplicating a subset of LLM weights (e.g., Attention or FFN) to serve as a dedicated vision branch. Conversely, models like **EVE** and **ELVA** integrate visual processing directly into the LLM backbone: EVE employs a sophisticated Patch Embedding layer (Convolution + Attention + Linear) for comprehensive pre-encoding, while ELVA interleaves a parameter-free Hierarchical Token Merging module within the LLM layers to effectively compress video tokens. These diverse approaches **reflect the community's effort to validate the potential of encoder-free LMMs through distinct architectural designs**.
>
> Similarly, we initially propose the **parameter-free Hierarchical Geometry Aggregation (HGA)** strategy within the LLM. This approach injects 3D inductive bias and compresses point tokens in a lightweight and effective manner. During this process, we identify a lack of sufficient interaction within point groups and introduce gated self-attention layer to facilitate adaptive adjustments.
>
> To further investigate how the placement of the gated attention layer—and thus the visual processing architecture—affects performance, we conduct the following ablation studies:
>
> **(1) Decoupling from the LLM Backbone (Integration into Token Embedding):**
> We remove the gated attention layer from the LLM backbone and relocate it to the Vision Part (Token Embedding layer) in two configurations:
> * **(a) Interleaved:** Inserting a gated attention layer into each stage of the point embedding.
> * **(b) Sequential:** Stacking the gated attention layer *after* the point embedding layer to function as a standalone visual feature extractor.
>
> **(2) Decoupled Attention Strategy (Adopting EVEv2/Mono-InternVL Paradigm):**
> To evaluate whether the architectural logic of EVEv2 and Mono-InternVL is transferable to the 3D domain, we modify the LLM layers such that visual and text tokens continue to share the FFN layer, but utilize **decoupled Attention layers**. In this setup, we deploy the gated attention layer to serve specifically as the Attention for point cloud tokens to verify its effectiveness.
>
> | Model / Configuration | Cap |  | Cls |QA |
> |-------|-----|-----|-----|---|
> |  | **GPT-4** | **Sentence-BERT** | **GPT-4** |**GPT-4** |
> | PointLLM-7B | 44.85 | 47.47 | 53.00 | 41.20 |
> | **ENEL-7B** | **51.03** | **48.79** | **55.55** | **43.80** |
> | +Interleaved in Token Emb. | 49.32 | 49.16 | 54.00 | 42.90 |
> | +Sequential Standalone | 49.61 | 49.32 | 54.00 | 42.70 |
> | +Decoupled Attention Strategy | 48.17 | 48.24 | 52.00 | 42.00 |
>
> For **Configs 1a & 1b:** Both strategies of moving the gated attention layer to the visual part, Point Embedding layer (whether interleaved or sequential), result in a noticeable performance drop compared to the default ENEL. For instance, the GPT-4 Captioning score decreases from **51.03%** to **49.32%** and **49.61%**, respectively. Placing the gated attention layer *within* the LLM backbone allows  to leverage the high-level semantic context extracted by preceding LLM layers to guide geometric feature aggregation. Restricting this process to the shallow "Vision Part" limits the semantic depth of the visual tokens.
>
> For **Config 2:** Adopting the decoupled Attention design (similar to EVEv2/Mono-InternVL) yields the lowest performance across most metrics (e.g., Classification drops to **52.50%**).
> While effective for dense 2D data with heavy vision weights, this decoupled approach appears less suitable for sparse 3D point clouds in an encoder-free setting. Our findings suggest that enabling full interaction between visual and textual information within shared Attention layers facilitates a more coherent alignment of geometry and language.
>
> We sincerely thank the reviewer for proposing these valuable architectural comparisons. We hope this detailed analysis and the results in **Appendix A.3.8 of main text PDF** fully address your concerns regarding the architectural design. We respectfully request that you reconsider the score based on these clarifications and the improved solidity of our work.

---

> ### Author Response · Authors · 2025-11-29
>
> Dear Reviewer fG4v,
>
> We sincerely appreciate the time and effort you have dedicated to our manuscript. Although the discussion phase is coming to an end, your insightful suggestions have significantly improved our work.
> **We are also grateful for your acknowledgment that your concerns have been addressed.**
>
> Sincerely,
>
> All authors of Submission 23290

---

### Official Review · Reviewer_FUhe · 2025-10-31

**Soundness:** 2
**Presentation:** 2
**Contribution:** 3
**Rating:** 4
**Confidence:** 4

**Summary:**

This paper presents the systematic investigation into encoder-free architectures for 3D LMMs. The authors propose two methods: (1) LLM-embedded Semantic Encoding with a novel Hybrid Semantic Loss during pre-training to capture high-level 3D semantics, and (2) Hierarchical Geometry Aggregation during instruction tuning to enable better perception of local geometric structures. The resulting model, ENEL-7B, achieves competitive performance with state-of-the-art encoder-based models on classification, captioning, and VQA tasks, demonstrating the viability of encoder-free architectures for 3D understanding.

**Strengths:**

1. The paper conducts extensive ablation studies comparing various encoder-free strategies, including different self-supervised losses and architectural components. This thorough investigation provides valuable insights and practical guidance for the community when designing 3D LMMs.

2. The experiments cover multiple benchmarks across different tasks, with both GPT-4 evaluation and traditional metrics reported.

**Weaknesses:**

1. All core investigations are built on PointLLM (Aug 2023), a 2-year-old model, while conveniently citing ShapeLLM (2024) only in Table 5. This is fundamentally problematic—the rapid evolution of LLMs makes findings from ancient baselines nearly irrelevant. The authors fail to justify why encoder-free strategies weren't explored on ShapeLLM, raising serious questions about whether these "insights" generalize at all or merely exploit weaknesses of an obsolete architecture.

2. Figure 3 is confusing and poorly designed. The distinctions between loss functions are unclear, undermining a key contribution of the paper.

3. When finally tested with modern components (Qwen2.5-7B + ShapeLLM data), ENEL-7B* achieves only 2% gain over ShapeLLM-13B on QA. This trivial improvement seriously undermines the paper's central claim that encoder-free is a promising direction—it suggests the approach offers minimal benefit as base models improve, questioning the entire motivation.

4. Including BLEU/ROUGE/METEOR in Table 5 directly contradicts ShapeLLM's findings (and the authors' own admission in A.3.2) that these metrics are meaningless for 3D LMMs.

5. The paper claims to be "the first comprehensive investigation" yet provides no related work discussion in the main paper (only in Appendix A.1). Reviewers are not obligated to read Appendix, and without proper context in the main text, it is impossible to assess the actual novelty and how this work differs from prior encoder-free efforts in 2D LMMs or recent 3D LMMs.

6. Table 5 omits many recent 3D LMMs from 2024-2025 (e.g., MiniGPT-3D, GreenPLM, and others). Without related work in the main paper, it's unclear whether this is due to incomplete survey or intentional cherry-picking. This makes it impossible to fairly assess ENEL's standing relative to the current state-of-the-art.

**Questions:**

1. Why not leverage the true advantage of encoder-free during training? The paper claims encoder-free architectures can "adapt to varying point cloud resolutions" (a key motivation in Abstract and Figure 1a). Can you train ENEL with mixed resolutions (e.g., randomly sample 2K-16K points per batch)? This would be infeasible for encoder-based models but natural for encoder-free models.

2. Can ENEL scale to scene-level understanding? All experiments are conducted on object-level point clouds (Objaverse, ModelNet40). Does ENEL maintain its advantages when handling much larger scene-level point clouds (e.g., ScanNet, 3D-GRAND)? This is crucial to assess whether encoder-free architectures can serve as a general solution for 3D understanding beyond isolated objects.

I appreciate the systematic investigation approach, but the evaluation does not sufficiently demonstrate the practical advantages of encoder-free design nor compare against modern baselines. If the authors can address these concerns, I would be willing to reconsider my score.

---

> ### Author Response · Authors · 2025-11-20
>
> > **W1: More Advanced Baseline Models.**
>
> Thank you for pointing this out.
> We chose PointLLM as the base model for exploring encoder-free architectures because:
>
> 1. **As a representative model in the 3D LMM field**, it has a sufficiently simple structure and an efficient training pipeline, making it an ideal foundation for validating feasibility.
>
> 2. From PointLLM to more recent 3D LMMs, the **high-level architecture largely follows the same pattern**: *encoder → projection → LLM*.
>    The primary differences lie in more diverse training data, stronger encoders and LLMs, and additional training stages.
>    This suggests that the insights drawn from ENEL are **likely to generalize across existing 3D LMMs**.
>
> To further verify this, we provide results based on **ShapeLLM** as the foundation model. As shown below, the encoder-free architecture offers **consistent performance improvements** across various tasks. For instance, on the 7B scale, we achieve a +6.34 increase in Captioning (Cap) task. The ablations performed on the ShapeLLM baseline further verify that both the Hybrid Semantic Loss and Hierarchical Geometry Aggregation strategy are essential.
>
> | Model | Cap |  | Cls |QA |
> |-------|-----|-----|-----|-----------|
> |  | **GPT-4** | **Sentence-BERT** | **GPT-4** | **GPT-4** |
> | ShapeLLM-7B | 46.92 | 48.20 | 54.50 | 47.40 |
> | ShapeLLM-13B | 48.94 | 48.52 | 54.00 | 53.10 |
> | ENEL-7B† | 53.26 | 48.75 | 56.00 | 48.90 |
> | ENEL-13B† | 54.78 | 49.37 | 56.00 | 54.80 |
> | - Hierarchical Geometry Aggregation | 50.83 | 48.16 | 52.50 | 52.20 |
> | - Hybrid Semantic Loss | 50.29 | 48.31 | 53.00 | 51.70 |
>
> **Note:** † indicates models built on ShapeLLM baseline.
>
> ---
> > **W2: Improved Loss Representation**
>
> Sorry for the confusion. We have **updated a clearer version of Figure 3 in the main paper PDF** to better illustrate the differences between various loss functions.
>
> Due to space limitations in the main paper, we present the mathematical formulation of Hybrid Semantic Loss in the appendix. To better highlight our contribution, we now **provide the mathematical formulation together with the motivation behind Hybrid Semantic Loss**.
>
> We observe that KD loss and contrastive learning loss require additional modules or larger batch sizes, yet achieve comparable or worse performance than pure data modeling losses (masked modeling and reconstruction). We propose Hybrid Semantic Loss by **exploiting two key properties**:
>
> 1. **Permutation invariance of point clouds**, allowing learnable tokens to be appended after visible tokens without positional restoration.
> 2. **The encoder-free architecture**, where 3D tokens are integrated into a causally masked LLM instead of a bidirectionally masked 3D encoder, fundamentally altering the information flow between visible and masked tokens. This allows visible tokens to learn harder objectives, such as spatial reconstruction, while learnable tokens focus on lightweight semantic understanding.
>
> Given an input point cloud $\{P _ i\}$, we partition it into $M$ point patches $\{G _ i\} \in \mathbb{R}^{M \times k \times 3}$ using the token embedding module, and obtain corresponding point tokens $\{F _ i\} \in \mathbb{R}^{M \times D}$.
> With masking ratio $r$, we separate tokens into masked $F _ {\text{mask}}$ and visible $F_{\text{vis}}$, with corresponding geometry patches $G_{\text{mask}}, G_{\text{vis}}$. We append learnable tokens $F_{\text{learn}}$ to maintain causal order, then jointly optimize:
>
> $$
> \mathcal{L} _ {\text{hybrid}} = \frac{1}{|F _ {\text{mask}}|} \|\hat{F} _ {\text{mask}} - F _ {\text{mask}}\| _ 2^2 + \frac{1}{|G _ {\text{vis}}|} \left(\sum _ {p \in \hat{G} _ {\text{vis}}} \min _ {q \in G _ {\text{vis}}} \|p - q\| _ 2^2 + \sum _ {q \in G _ {\text{vis}}} \min _ {p \in \hat{G} _ {\text{vis}}} \|q - p\| _ 2^2\right)
> $$
> where the first term is the semantic term enforcing token-level feature prediction, and the second term is the geometric term ensuring spatial structure preservation.

---

> ### Author Response · Authors · 2025-11-20
>
> > **W3: Effectiveness of LLM Base Model.**
>
> While the absolute QA gain over ShapeLLM-13B is around 2 points in that specific setting, we emphasize that this is achieved with a 7B encoder-free model, which is 46% smaller parameter size. We adopt Qwen2.5-14B to unify the model size. ENEL-14B* achieves GPT-4 scores that are 11.79 points, 7.5 points, and 9.1points higher than ShapeLLM-13B on Captioning, Classification, and QA tasks, respectively. Compared to ShapeLLM-7B, our ENEL-7B* achieves a 10.99 points, 6.5 points and 7.8 points improvement on Captioning, Classification and QA tasks.
>
> | Model | Cap |  | Cls |QA |
> |-------|-----|-----|-----|-----------|
> |  | **GPT-4** | **Sentence-BERT** | **GPT-4** | **GPT-4** |
> | ShapeLLM-7B | 46.92 | 48.20 | 54.50 | 47.40 |
> | ShapeLLM-13B | 48.94 | 48.52 | 54.00 | 53.10 |
> | ENEL-7B* | 57.91 | 49.90 | 61.00 | 55.20 |
> | ENEL-14B* | 60.73 | 51.32 | 61.50 | 62.20 |
>
> For fair comparison and to validate the effectiveness of the LLM base model, we systematically choose ShapeLLM as the baseline model for verification, rather than PointLLM. As shown in the table below, **switching the base model from Vicuna V1.1 to Qwen2.5 yields an improvement of 8.4 points on QA Task**.
>
> | Model | Cap |  | Cls |QA |
> |-------|-----|-----|-----|-----------|
> |  | **GPT-4** | **Sentence-BERT** | **GPT-4** | **GPT-4** |
> | ShapeLLM-7B | 46.92 | 48.20 | 54.50 | 47.40 |
> | ENEL-ShapeLLM-7B | 53.26 | 48.75 | 56.00 | 48.90 |
> | ENEL-ShapeLLM-Qwen2.5-7B | 58.59 | 49.87 | 61.50 | 57.30 |
>
> ---
> > **W4: BLEU/ROUGE/METEOR Metrics.**
>
> Thank you for pointing this out. These metrics are indeed not meaningful. We chose to use them to **align with PointLLM and previous 3D LMMs for comparison purposes**. We have de-emphasized these metrics in the PDF to highlight the importance of other metrics such as GPT-4 score.
>
> ---
> > **W5: Related Work.**
>
> Due to space limitations, we had to place part of the preliminary knowledge in Section 3.1 of the main text, while the related work was placed in the appendix. We have added the related work to the Section 2 of the main text, with particular emphasis on **3D LMMs** and **encoder-free 2D LMMs**, as well as the **distinctions between ENEL (the first exploration of encoder-free 3D LMM) and both lines of work**.
>
> ---
> > **W6: More Comparisons.**
>
> We have now included MiniGPT-3D and GreenPLM in the Objaverse benchmark, with reproduced results clearly marked using $^\alpha$. Since MiniGPT-3D adopts an extended four-stage training pipeline based on PointLLM data and uses a 2D LMM as its base model, we extend from two-stage to four-stage training for fair comparison, denoted with *. Results on the Objaverse benchmark show that under comparable training settings, ENEL consistently outperforms these methods across all metrics.
>
> | Model | Cap |  | |Cls |
> |-------|-----|-----|-----|-----------|
> |  | **GPT-4** | **Sentence-BERT** | **SimCSE** | **GPT-4** |
> | PointLLM-7B | 44.85 | 47.47 | 48.55 | 53.00 |
> | PointLLM-7B* | 48.97 | 48.23 | 49.02 | 55.00 |
> | PointLLM-13B | 48.15 | 47.91 | 49.12 | 54.00 |
> | PointLLM-13B* | 52.26 | 48.78 | 50.10 | 56.50 |
> | MiniGPT-3D$^\alpha$ | 52.49 | 48.73 | 49.26 | 54.50 |
> | GreenPLM$^\alpha$ | 47.78 | 48.30 | 48.53 | 53.50 |
> | ENEL-7B | 51.03 | 48.79 | 49.52 | 55.55 |
> | ENEL-7B* | 54.56 | 49.89 | 50.61 | 58.00 |
> | ENEL-13B | 53.24 | 48.92 | 50.17 | 56.00 |
> | **ENEL-13B*** | **57.93** | **50.12** | **51.30** | **60.00** |
>
> Note: $^\alpha$ denotes reproduced results. * denotes extended four-stage training.

---

> ### Author Response · Authors · 2025-11-20
>
> > **Q1: Train with Mixed Resolutions.**
>
> Thank you for pointing this out. The encoder-free architecture is better suited for training with mixed resolutions compared to encoder-based 3D LMMs. To validate its effectiveness, we randomly sample 2K–16K points per batch during the pre-training stage and evaluate performance at different resolutions on the Objaverse Captioning task using GPT-4 for scoring.
>
> As shown in the table, ENEL-7B substantially improves robustness over PointLLM-7B by reducing performance variance (4.56 → 2.38) and lowering the relative drop between the best and worst resolutions (25.45% → 12.55%). **ENEL-Mix-7B further enhances stability, achieving the lowest standard deviation (2.08) and smallest relative drop (10.40%)**. In terms of overall accuracy, ENEL-7B increases the average score from 38.72 to 48.18 (+9.46), with **ENEL-Mix-7B reaching 49.46**.
>
> | Method | 2K/128 | 4K/256 | 8K/512 | 12K/1024 | 16K/2048 |
> |--------|--------|--------|--------|----------|----------|
> | PointLLM-7B | 33.7 | 41.4 | 44.0 | 41.7 | 32.8 |
> | ENEL-7B | 44.6 | 49.0 | 51.0 | 50.0 | 46.3 |
> | ENEL-Mix-7B | 46.5 | 50.4 | 51.9 | 50.8 | 47.7 |
>
> We also evaluate ENEL-Mix-7B across multiple tasks, demonstrating that mixed-resolution training further enhances the effectiveness of the encoder-free architecture.
>
> | Model | Cap |  | Cls |QA |
> |-------|-----|-----|-----|---|
> |  | **GPT-4** | **Sentence-BERT** | **GPT-4** |**GPT-4** |
> | PointLLM-7B | 44.85 | 47.47 | 53.00 | 41.20 |
> | ShapeLLM-7B | 46.92 | 48.20 | 54.50 | 47.40 |
> | ENEL-7B | 51.03 | 48.79 | 55.55 | 43.80 |
> | ENEL-Mix-7B | 51.92 | 49.06 | 56.00 | 44.40 |
>
> ---
> > **Q2: Scene-Level Experiments.**
>
> Although our method is initially designed for object-level understanding, ENEL can also be applied to scene-level scenarios in a zero-shot manner without further training. We use a parameter-free relational structure to model the objects segmented from the scene and concatenate them with text tokens as input to the LLM. We perform experiments on the ScanQA and Scan2Cap benchmarks.
>
> | Method | ScanQA |  |  | Scan2Cap\@0.25 |  |  |
> |--------|--------|--------|--------|--------|--------|--------|
> |  | **METEOR** | **ROUGE** | **CIDER** | **METEOR** | **ROUGE** | **CIDER** |
> | Qwen2VL-7B | 11.4 | 29.3 | **53.9** | 16.7 | 24.7 | 0.0 |
> | PointLLM-7B | 10.2 | 23.9 | 43.5 | 13.1 | 28.8 | 30.9 |
> | ShapeLLM-7B | 12.3 | 26.8 | 49.0 | 16.0 | 32.2 | 33.4 |
> | ENEL-7B | **13.5** | **31.2** | 52.7 | **18.4** | **34.5** | **35.6** |
>
> To further validate its effectiveness, we add a dedicated scene-level training stage after the pre-training and SFT stages. We extract 110K samples of 3D captioning and 3D QA data from the LEO-Instruct dataset[1] as training data for this third stage. The model architecture remains the same as in the zero-shot setting. Our results on the ScanQA and Scan2Cap benchmarks demonstrate performance **comparable to current 3D LMMs that take 3D scene point clouds as input**.
>
> | Method | ScanQA |  |  | Scan2Cap\@0.25 |  |  |
> |--------|--------|--------|--------|--------|--------|--------|
> |  | **METEOR** | **ROUGE** | **CIDER** | **METEOR** | **ROUGE** | **CIDER** |
> | LL3DA[2] | 15.9 | 37.3 | 76.8 | 26.0 | 55.1 | 65.2 |
> | ChatScene[3] | 18.0 | 41.6 | 87.7 | **28.0** | **58.1** | 77.2 |
> | 3D-LLaVA[4] | **18.4** | 43.1 | 92.6 | 27.1 | 57.7 | **78.8** |
> | ENEL-Scene-7B | 17.8 | **44.2** | **95.0** | 27.4 | 56.9 | 72.4 |
>
> [1] Huang, et al. "An embodied generalist agent in 3d world." arXiv preprint arXiv:2311.12871.
>
> [2] Chen, et al. "Ll3da: Visual interactive instruction tuning for omni-3d understanding reasoning and planning." CVPR 2024.
>
> [3] Zhang, et al. "Chatscene: Knowledge-enabled safety-critical scenario generation for autonomous vehicles." CVPR 2024.
>
> [4] Deng, et al. "3d-llava: Towards generalist 3d lmms with omni superpoint transformer." CVPR 2025.

---

> ### Author Response · Authors · 2025-11-25
> **Summary of PDF Revisions**
>
> We sincerely appreciate the reviewer’s insightful feedback and remain committed to strengthening the paper. The following key revisions have been incorporated:
>
> * **Section 2:** Added a **Related Work** section to discuss 3D LLMs and 2D Encoder-free LMMs, explicitly clarifying the distinctions between ENEL and these approaches.
> * **Figure 3:** Revised the figure to present a more intuitive comparison of the loss functions.
> * **Figure 5:** Incorporated **MiniGPT-3D** comparison metrics and visually de-emphasized the BLEU/ROUGE/METEOR metrics to highlight key performance indicators.
> * **Appendix A.3.6:** Added detailed **Ablation Studies** covering:
>     * Robustness to Resolution Variations.
>     * Extension to Scene-Level Understanding.
>     * Generalizability across Baselines.
>
> ---
>
> These updates are highlighted in **blue** for easy reference.

---

> ### Author Response · Authors · 2025-11-27
> **Conclusion of Rebuttal and Invitation for Re-evaluation**
>
> Dear Reviewer FUhe,
>
> We remain highly appreciative of your time, effort, and insightful critiques on our manuscript.
>
> We have incorporated comprehensive new experiments and detailed analyses, which significantly strengthen the core claims of our paper. As the rebuttal phase concludes, we kindly invite your attention to these updates to confirm that all raised concerns have been comprehensively addressed.
>
> We hope that these improvements warrant a positive re-evaluation of our work. Should any residual issues remain, we are prepared to provide any further comprehensive experiments.
>
> Best regards,
>
> Paper 23290 Authors

---

> > ### Comment · Reviewer_FUhe · 2025-11-27
> > **Post-rebuttal**
> >
> > I would appreciate it if the authors could include the results of ENEL-ShapeLLM-7B in Table 5. Also, please include this important comparison for base models in the Appendix.
> >
> > Most of my concerns have been addressed. I will raise my score.
> >
> > Note: I fully understand that supplementary experiments take time, and I acknowledge the authors' efforts. However, please submit the entire rebuttal in a single update. Reviewers do not wish to receive multiple notifications from partial updates.

---

> > > ### Author Response · Authors · 2025-11-27
> > >
> > > We express our sincere gratitude for your positive re-evaluation and the decision to raise the score. We strictly follow your final suggestions and have incorporated the following updates into the revised manuscript:
> > >
> > > * **Update to Table 5:** We have included the results of **ENEL-ShapeLLM-7B** and **ENEL-ShapeLLM-13B** in Table 5 to provide a more comprehensive comparison.
> > > * **Base Model Analysis:** We have added the comparison of base models in **Table 11 of Appendix**.
> > >
> > > We also appreciate your reminder regarding the submission process. We have consolidated all these supplementary results into the final revised PDF to ensure a streamlined update.
> > >
> > > Thank you again for your time and constructive guidance throughout the review process.

---

> ### Author Response · Authors · 2025-11-29
>
> Dear Reviewer FUhe,
>
> We sincerely appreciate the time and effort you have dedicated to our manuscript. Although the discussion phase is coming to an end, your insightful suggestions have significantly improved our work.
> **We are also grateful for your acknowledgment that your concerns have been addressed and for your willingness to consider raising the score.**
>
> Sincerely,
>
> All authors of Submission 23290

---

### Official Review · Reviewer_t3wM · 2025-11-01

**Soundness:** 2
**Presentation:** 2
**Contribution:** 2
**Rating:** 6
**Confidence:** 3

**Summary:**

Towards 3D understanding using LLM architecture, this work proposed an encoder-free method that works well on classification, captioning and QA tasks. First, during pre-training stage, multiple self-supervised losses are tried and compared. Second, at the instruction-tuning stage, geometry aggregation and propagation operations are baked into the LLM layers to better use the local details of 3D point clouds.

**Strengths:**

1. It is interesting to consider encode-free alternative for 3D LLM solutions. It has the potential to overcome the restricted resolution issue, and to reduce the semantic gaps between encoder and LLM.
2. The proposed approach achieves good performance on 3D benchmarks and sometimes is state of the art.

**Weaknesses:**

1. The approach is quite incremental in terms of contributions. 1) For hybrid loss choices, combining masked and reconstruction loss seems less exciting to me. 2) The geometry aggregation is designed specifically for 3D point clouds. In some way, it moves the functionality of 3D point encoder inside the LLM. Although it might work, this leads to the loss of generality of using an general-purpose LLM.

2.  Back to the claims of the paper. Two limitations motivate this work. 1) The variable resolution. However, there's no results to show that the method works for the case, and how does it work. Maybe benchmarks for this is missing. 2) The semantic gap between encoder and LLM. I would like to see some evidence of this first. Then, I would like to see some results of this work alleviating the issue.

**Questions:**

1. What are other models that you could use as baseline except `PointLLM-7B`?
2. What is the performance of the proposed method using following ablations? The reasoning behind this is to see how the vanilla version without the `claimed contributions` works.
- a. without geometry aggregation, eg. $l=0$
- b. the variant with $H=0$

---

> ### Author Response · Authors · 2025-11-19
>
> > **W1: Explanation of Approach Innovation.**
>
> We present an **empirical study** exploring the potential of encoder-free architectures in 3D LMMs. The core idea is to transfer the functionality of the 3D encoder to the LLM rather than focusing on designing novel modules. To this end, we propose (1) Hybrid Semantic Loss, which reveals how different self-supervised objectives **interact with the encoder-free 3D tokenization mechanism within the LLM**; and (2) Hierarchical Geometry Aggregation Strategy, which **injects lightweight yet effective 3D inductive biases into the LLM** without compromising its general language capabilities.
>
> • Hybrid Semantic Loss
>
> The loss is not a simple combination of masked and reconstruction losses. Our motivation stems from the observation that KD loss and contrastive learning loss require additional modules or larger batch sizes, yet perform comparably or worse than pure data modeling losses (masked modeling and reconstruction). The innovation lies in **addressing the structural mismatch between 3D data and LLMs**. We propose Hybrid Semantic Loss by exploiting two key properties: (1) **the permutation invariance of point clouds**, allowing learnable tokens to be appended after visible tokens without positional restoration; and (2) **the encoder-free architecture**, where 3D tokens are integrated into a causally-masked LLM instead of a bidirectionally-masked 3D encoder, fundamentally altering information flow between visible and masked tokens, enabling visible tokens to learn harder objectives while learnable tokens focus on lightweight reconstruction.
>
> • Hierarchical Geometry Aggregation Strategy
>
> The strategy transfers the specific domain knowledge of 3D encoders (global-to-local hierarchy) into the LLM, effectively compressing point cloud visual tokens and extracting local information using 3D-domain methods. Since this **operates solely on the visual side**, it does not affect text-side processing or performance.
>
> The decline in LLM generalization likely **stems from catastrophic forgetting** caused by using only multimodal data in both training stages without pure text data—a phenomenon commonly observed in 2D LMMs (e.g., LLaVA). As shown in the table below, we provide experimental validation on the MMLU Benchmark[1], where we evaluate the general language capabilities of ENEL, ENEL after removing the Hierarchical Geometry Aggregation Strategy, and ENEL trained with an additional 12K sampled pure text data from Evol-Instruct-GPT4-Turbo-143K[2] in the SFT stage.
>
> | Method | MMLU (5-shot) |
> |--|--|
> | Vicuna-7B | 47.1 |
> | ENEL-7B | 46.4 |
> | - HGA | 46.5 |
> | + Text Data | 47.3 |
>
> ---
> > **W2.1: Experiments of the variable resolution.**
>
> Sorry for the confusion. In Figure 1(a) of the main text, we present a performance comparison between ENEL and the baseline model PointLLM across different point cloud resolutions on the Objaverse Captioning task. The results can be summarized in four key points:
>
> **1. Lower Performance Variance.** ENEL exhibits substantially lower variability across the five resolutions, with a standard deviation of 2.38 compared to 4.56 for PointLLM, indicating more stable performance.
>
> **2. Smaller Relative Drop.** The relative performance drop from the best to worst resolution is only 12.55% for ENEL, whereas PointLLM suffers a 25.45% drop, demonstrating reduced sensitivity to resolution changes.
>
> **3. Higher Average Performance.** ENEL consistently outperforms PointLLM across all resolutions, achieving an average score of 48.18 versus 38.72, achieving a 9.46-point advantage.
>
> **4. Stronger Worst-Case Robustness.** Even in the worst case, ENEL attains 44.6, which is 11.8 points higher than PointLLM's minimum of 32.8, maintaining reliability even under suboptimal resolutions.
>
> We **attribute this robustness to the encoder-free architecture**, which removes the dependency on a fixed-resolution pre-trained encoder and allows the LLM to directly adapt to varying point counts.
>
> To further validate the effectiveness of the encoder-free architecture in alleviating this limitation, we conduct an additional experiment during the pretraining stage by **mixing resolutions**, randomly sampling 2K–16K points per batch—a capability that is non-trivial or inefficient for the original encoder-based 3D LMMs, which are typically pre-trained at a fixed resolution. The results are validated on the Objaverse Captioning task across different resolutions.
>
> ENEL-Mix-7B further improves robustness: its standard deviation drops from 2.38 to 2.08, and the relative performance drop decreases from 12.55% to 10.40%.
>
> | Method | 2K/128 | 4K/256 | 8K/512 | 12K/1024 | 16K/2048 |
> |-|-|-|-|-|-|
> | PointLLM-7B | 33.7 | 41.4 | 44.0 | 41.7 | 32.8 |
> | ENEL-7B | 44.6 | 49.0 | 51.0 | 50.0 | 46.3 |
> | ENEL-Mix-7B | 46.5 | 50.4 | 51.9 | 50.8 | 47.7 |
>
> [1] "Measuring massive multitask language understanding."
>
> [2]Chen, et al. "Allava: Harnessing gpt4v-synthesized data for lite vision-language models."

---

> ### Author Response · Authors · 2025-11-19
>
> > **W2.2: Visual Evidence of Semantic Gap.**
>
> We visualize the attention scores of the average text token to the point tokens at the second-to-last layer of the LLM (Figure 1(b) in main text) to provide qualitative proof.
>
> **1.Evidence of the Gap**: In encoder-based models (e.g., PointLLM), we observe that the **attention to visual tokens is scattered and unfocused**, as the simple projection layer cannot effectively perform semantic transformation, and the self-supervised objective of the 3D encoder is not aligned with the needs of the LLM.
>
> **2.Evidence of Alleviation**: In contrast, ENEL exhibits **highly concentrated attention on semantically relevant regions**. Since our visual tokens are trained end-to-end with the LLM from scratch (without a misaligned encoder), the "visual vocabulary" is inherently learned in the textual semantic space. This allows the LLM to precisely attend to fine-grained details.
>
> ---
> > **Q1: More Baseline Models.**
>
> Thank you for pointing this out. The encoder-free architecture **can be applied to existing encoder-based 3D LMMs** for experimentation. We use ShapeLLM as the baseline model to provide supplementary experiments, where we maintain the same LLM, training data, and training settings. As shown in the table below, we demonstrate **the effectiveness of the encoder-free architecture across multiple benchmarks** and validate that Hybrid Semantic Loss and Hierarchical Geometry Aggregation Strategy are critical components.
>
> | Model | Cap |  | Cls |QA |
> |-------|-----|-----|-----|-----------|
> |  | **GPT-4** | **Sentence-BERT** | **GPT-4** | **GPT-4** |
> | ShapeLLM-7B | 46.92 | 48.20 | 54.50 | 47.40 |
> | ShapeLLM-13B | 48.94 | 48.52 | 54.00 | 53.10 |
> | ENEL-7B† | 53.26 | 48.75 | 56.00 | 48.90 |
> | ENEL-13B† | 54.78 | 49.37 | 56.00 | 54.80 |
> | - Hierarchical Geometry Aggregation | 50.83 | 48.16 | 52.50 | 52.20 |
> | - Hybrid Semantic Loss | 50.29 | 48.31 | 53.00 | 51.70 |
>
> **Note:** † indicates models built on ShapeLLM baseline.
>
> ---
> > **Q2: More Ablation Study.**
>
> In Table 3 of the main text, we present the experimental results for *l*=0. In the table below, we provide more comprehensive metrics when removing the Hierarchical Geometry Aggregation strategy, **validating its necessity**. Meanwhile, the corresponding extended results of **H = 0** variant  are shown in the table below. These findings indicate that interleaving LLM layers after geometric aggregation layers enables the model to refine semantic understanding alongside geometric compression, outperforming the approach of performing all geometric compression and propagation in a single continuous step.
>
> | Model | Cap |  | Cls | QA |
> |-------|-----|-----|-----|---|
> |  | **GPT-4** | **Sentence-BERT** | **GPT-4** |**GPT-4** |
> | ENEL-7B | 51.03 | 48.79 | 55.55 | 43.80 |
> | *l* = 0 | 47.65 | 47.30 | 52.00 | 41.20|
> | *H* = 0 | 49.83 | 48.34 | 53.50 | 42.00|

---

> ### Author Response · Authors · 2025-11-25
> **Summary of PDF Revisions**
>
> We sincerely appreciate the reviewer’s insightful feedback and remain committed to strengthening the paper. The following key revisions have been incorporated:
>
> * **Highlighted empirical evidence** for the two identified limitations in the Figure Caption of **Figure 1**.
> * **Added the motivation** for the Hybrid Semantic Loss at the end of **Section 3.2**.
> * **Included detailed Ablation Studies** in **Appendix A.3.6** covering:
>     * General Language Capabilities.
>     * Generalizability across Baselines.
>     * Impact of Hierarchical Aggregation Design.
>
> ---
>
> These updates are highlighted in **blue** for easy reference.

---

> ### Author Response · Authors · 2025-11-27
> **Conclusion of Rebuttal and Invitation for Re-evaluation**
>
> Dear Reviewer t3wM,
>
> We remain highly appreciative of your time, effort, and insightful critiques on our manuscript.
>
> We have incorporated comprehensive new experiments and detailed analyses, which significantly strengthen the core claims of our paper. As the rebuttal phase concludes, we kindly invite your attention to these updates to confirm that all raised concerns have been comprehensively addressed.
>
> We hope that these improvements warrant a positive re-evaluation of our work. Should any residual issues remain, we are prepared to provide any further comprehensive experiments.
>
> Best regards,
>
> Paper 23290 Authors

---

> ### Author Response · Authors · 2025-11-28
> **Follow-up on Rebuttal: Discussion Phase Closing Soon**
>
> **Dear Reviewer t3wM**,
>
> We sincerely appreciate your valuable comments, which have greatly contributed to improving our work.
>
> Following your suggestions, we have **revised Table 1 to provide empirical evidence for the two motivations**. In **Appendix A3.6**, we further elaborate on the exploration of **general text understanding ability, the extended set of baseline models, and the ablation study of the Hierarchical Geometry Aggregation design**.
>
> We hope that these revisions and clarifications have addressed your concerns. If you find our responses satisfactory, we would be grateful if you could kindly reconsider our score. Otherwise, please feel free to let us know if you have any further questions—we would be more than happy to provide additional clarification.
>
> Thank you again for your insightful comments, which have helped us enhance the clarity and completeness of our work.
>
> Sincerely,
>
> All authors of Submission 23290

---

> ### Author Response · Authors · 2025-11-29
>
> Dear Reviewer t3wM,
>
> We sincerely appreciate the time and effort you have dedicated to our manuscript. Although the discussion phase is coming to an end, your insightful suggestions have significantly improved our work.
>
> Sincerely,
>
> All authors of Submission 23290

---

### Author Response · Authors · 2025-11-25
**General Response**

**Dear Reviewers, Area Chairs, and Senior Area Chairs,**

We sincerely appreciate the time and effort you have dedicated to reviewing our paper.

---
We are grateful for the reviewers' recognition of our work as **the first comprehensive investigation into the potential of encoder-free architectures for 3D Large Multimodal Models (LMMs)**. ENEL challenges the conventional reliance on heavy pre-trained 3D encoders, demonstrating that transferring the functionality of the encoder into the LLM (via our proposed *LLM-embedded Semantic Encoding* and *Hierarchical Geometry Aggregation*) can effectively alleviate long-standing resolution limitations and semantic discrepancies. We believe this work provides a solid foundation for simplifying 3D-LMM pipelines while achieving performance on par with, or even surpassing, state-of-the-art encoder-based models.

Overall, we are encouraged by the positive feedback highlighting:
* **The novelty and clear motivation of the encoder-free design**, which identifies and effectively alleviates key challenges such as **resolution mismatch** and **semantic misalignment** in traditional 3D LMMs (Reviewers **t3wM, fG4v**).
* **The effectiveness of the proposed self-supervised loss**, particularly the thorough exploration of objectives (e.g., masked modeling, reconstruction) that culminates in the **Hybrid Semantic Loss** for robust pre-training (Reviewers **fG4v, FUhe**).
* **The depth and value of the experimental analysis**, where extensive ablation studies on architectural components provide valuable insights and practical guidance for future research in the community (Reviewers **FUhe, fG4v**).
* **Strong empirical results and efficiency**, demonstrating **state-of-the-art** performance on multiple benchmarks (including GPT-4 evaluations) while offering notable **compute and memory savings** compared to strong baselines (Reviewers **t3wM, fG4v, FUhe**).

---
To address the reviewers' concerns, we have conducted several additional experiments and analyses, including:

* **Extended evaluation to scene-level and pure text tasks**, validating the effectiveness and generalization of the encoder-free architecture beyond object-level benchmarks (Reviewers **t3wM, FUhe**).
* **Conducted controlled experiments** to isolate the architectural advantages and demonstrate consistent performance gains of the encoder-free design across different baselines (Reviewers **t3wM, FUhe, fG4v**).
* **Validated the resolution motivation** across multiple tasks and introduced **mixed-resolution pre-training**, highlighting the unique flexibility and advantages of the encoder-free architecture (Reviewers **t3wM, FUhe, fG4v**).
* **Provided comprehensive efficiency and component analysis**, detailing comparisons between our Point Embedding/Aggregation modules and their counterparts in 2D encoder-free and 3D encoder-based LMMs (Reviewer **fG4v**).

---
**Main Text Revisions:**
* **Figure 1:** Updated the caption to highlight empirical evidence regarding the identified limitations.
* **Section 2 (Related Work):** Added a dedicated discussion on 3D LLMs and 2D Encoder-free LMMs, explicitly clarifying the distinctions between ENEL and these approaches.
* **Section 3.2:** Added the motivation for the Hybrid Semantic Loss at the end of the section.
* **Figure 3:** Revised the figure to present a more intuitive comparison of the loss functions.
* **Figure 5:** Incorporated **MiniGPT-3D** comparison metrics and visually de-emphasized BLEU/ROUGE/METEOR metrics to highlight key performance indicators.

**Appendix Revisions:**
* **Appendix A.3.5:** Formalized the strict definition of **"Encoder-free"** architecture and provided a comparative analysis against established 2D/Video LMM standards.
* **Appendix A.3.6:** Added comprehensive **Ablation Studies** covering:
    * Tokenizer Architecture and the Impact of Hierarchical Aggregation.
    * Hyperparameter Sensitivity Analysis.
    * Generalizability across Baselines.
    * General Language Capabilities and Extension to Scene-Level Understanding.
* **Appendix A.3.7:** Presented systematic **resolution sweep results** across different tasks, along with the **visualization** and discussion of specific failure modes.

---
We explicitly thank the reviewers and ACs for their thoughtful comments during the discussion period. We hope our responses have adequately addressed all concerns and welcome any further discussion. Thank you!

---

### Author Response · Authors · 2025-11-29
**Rebuttal Summary and Reviewer Score Updates**

**Dear Area Chairs and Senior Area Chairs,**

We sincerely appreciate your time and effort in reviewing our manuscript.

---
**1. Reviewer t3wM (Score: 6 $\to$ No Response)**

**Concerns:**
1.  **Novelty & Generality:** Questioned the novelty of the hybrid loss and whether geometry-specific designs compromise LLM generality.
2.  **Evidence for Motivations:** Noted a lack of empirical evidence supporting claims regarding variable resolution and the semantic gap.
3.  **Baseline Sufficiency:** Criticized the reliance on a single baseline (PointLLM-7B) and requested additional models.
4.  **Ablation Studies:** Requested specific ablations (e.g., removing geometry aggregation) to verify component effectiveness.

**Our Response:**
1.  **Clarified Novelty:** We clarified the Hybrid Semantic Loss innovation, added MMLU experiments, and confirmed that language degradation stems purely from absent text data.
2.  **Supported Motivations:** We highlighted evidence in Figure 1 and provided mixed-resolution training experiments to validate our claims.
3.  **Expanded Baselines:** We integrated **ShapeLLM** as an additional baseline to broaden the comparative analysis.
4.  **Conducted Ablations:** We provided the full set of requested ablation studies in **Appendix A.3.6**.

**Status:** No response received.

---
**2. Reviewer FUhe (Score: 4 $\to$ 6)**

**Concerns:**
1.  **Outdated Baseline:** Criticized the use of PointLLM (2023) as the primary baseline given the rapid evolution of LLMs.
2.  **Metrics & Structure:** Noted inconsistent metrics (BLEU/ROUGE) and the absence of a *Related Work* section in the main text.
3.  **Incomplete Comparisons:** Highlighted the omission of recent SOTA 3D LMMs (2024-2025).
4.  **Variable Resolution:** Pointed out a lack of experiments demonstrating the benefits of mixed-resolution training.
5.  **Scene-Level Generalization:** Questioned the model's ability to generalize to scene-level understanding.

**Our Response:**
1.  **Integrated ShapeLLM:** We validated our method using **ShapeLLM (2024)** in **Appendix A.3.6**, demonstrating consistent performance gains.
2.  **Refined Structure:** We deprioritized traditional metrics and added a dedicated **Related Work** section (Section 2).
3.  **Updated SOTA:** We expanded **Table 5** to include recent models like **MiniGPT-3D**, showing ENEL remains robust.
4.  **Verified Mixed-Resolution:** We provided mixed-resolution pre-training results in **Appendix A.3.6**, confirming enhanced robustness.
5.  **Validated Scene-Level:** We included zero-shot and fine-tuning results for scene-level tasks in **Appendix A.3.6**.

**Status:** Acknowledged that most concerns were addressed; Score raised to 6.

---
**3. Reviewer fG4v (Score: 4 $\to$ Addresses Most Concerns)**

**Concerns:**
1.  **"Encoder-Free" Definition:** Argued that the "encoder-free" claim was overstated.
2.  **Hyperparameter Justification:** Questioned the fixed loss coefficients and requested sensitivity analysis.
3.  **Fairness of Comparisons:** Criticized experimental settings as potentially unfair.
4.  **Tokenizer Necessity:** Requested comparisons against a minimal linear projection to justify the tokenizer's complexity.
5.  **Computational Breakdown:** Demanded a granular breakdown of FLOPs and latency (grid building/aggregation).
6.  **Resolution Analysis:** Noted the absence of systematic sweeps to validate robustness against variable resolutions.

**Our Response:**
1.  **Defined Terminology:** We formally defined "Encoder-free" in **Appendix A.3.5**, aligning strictly with 2D/Video domain standards.
2.  **Hyperparameter Analysis:** We conducted a detailed sensitivity analysis of loss coefficients in **Appendix A.3.6**.
3.  **Clarified Fairness:** We validated our experimental settings and confirmed that our comparisons are methodologically sound.
4.  **Tokenizer Ablation:** We validated the necessity and efficiency of the hierarchical FPS/k-NN structure via ablations in **Appendix A.3.6**.
5.  **Computed Costs:** We provided a granular, component-wise comparison of parameters and FLOPs against PointLLM.
6.  **Verified Robustness:** We reported comprehensive performance metrics across varying resolutions for multiple tasks in **Appendix A.3.7**.

**Status:** Acknowledged that most concerns were addressed.

---
**Final Score Summary**

| Reviewer | Initial | Final | Confidence | Engagement |
| -- | --- | --- | --- | ----- |
| **t3wM** | 6  | **6** | 3  | ❌ No response |
| **FUhe** | 4 | **6** | 4  | ✔️ Replied and increased score      |
| **fG4v** | 4  | **4** | 5  | ✔️ Replied and addressed most concerns, but the discussion phase concluded prematurely.  |

**Average score: 5.33 (min 4, max 6)**
**Two reviewers who initially provided lower scores engaged in the discussion: one subsequently raised their rating, while the other indicated that most concerns were resolved but could not continue due to the early termination of the discussion period.**

---

### Meta-Review · Area_Chair_dWT3 · 2025-12-10

**Summary:**

This paper proposes an encoder-free architecture for 3D large multimodal models (LMMs), introducing a hybrid semantic loss for pre-training and a hierarchical aggregation mechanism for instruction tuning. Experiments demonstrate improved performance over state-of-the-art 3D LLMs under same settings. A central concern across reviewers is the claim of being “encoder-free”: Reviewers FUhe and fG4v argued that the proposed design appears to relocate encoder functionality to the decoder, since hierarchical geometry aggregation still performs hierarchical feature aggregation similar to conventional 3D encoders. The authors clarified their definition of encoder-free architectures—consistent with prior encoder-free 2D VLMs—and addressed this concern. One reviewer was initially positive, and the other two indicated that most of their concerns were resolved and that they intend to raise their scores.

**Reviewer Concerns:**

The primary concern for all reviewers is the encoder-free claim, with FUhe and fG4v noting that the architecture still includes feature aggregation akin to a conventional 3D encoder. This was addressed through clarification regarding the design philosophy shared with prior 2D encoder-free VLMs. Other major concerns—such as the lack of comparison with stronger baselines (e.g., ShapeLLM), insufficient computation breakdown, and mixed-resolution training—were also addressed through added clarification and experiments.

**Reviewer Scores:**

Reviewer t3wM gave an initial score of 6, raising concerns about the novelty of the hybrid loss, limited empirical evidence supporting claims about variable resolution and semantic gaps, missing baselines, and insufficient ablations. Most concerns were addressed, and the reviewer is likely to maintain a score of 6. Reviewer FUhe gave an initial score of 4, citing concerns about the encoder-free claim, comparison with ShapeLLM, mixed-resolution training, scene-level generalization, and literature coverage. The rebuttal and new experiments resolved most issues, and the reviewer indicated they would raise the score to 6 if able to participate fully in the discussion. Reviewer fG4v also gave an initial score of 4, with concerns regarding the encoder-free claim, hyperparameter justification, computation breakdown, and resolution analysis. After clarification and added experiments, the reviewer confirmed that most concerns have been addressed and thus likewise intends to raise the score to 6.

---

### Decision · Program_Chairs · 2026-01-26

Accept (Poster)